

# Source Apportionment of VOCs, IVOCs, and SVOCs by Positive Matrix Factorization in Suburban Livermore, California

Rebecca A. Wernis[1,2], Nathan M. Kreisberg[3], Robert J. Weber[2], Greg T. Drozd[4], Allen H. Goldstein[1,2]

[1]Department of Civil and Environmental Engineering, University of California Berkeley, Berkeley, CA 94720, USA
[2]Department of Environmental Science, Policy and Management, Berkeley, CA 94720, USA
[3]Aerosol Dynamics, Inc., Berkeley, CA 94710, USA
[4]Department of Chemistry, Colby College, Waterville, ME 04901, USA

*Correspondence to*: Rebecca Wernis (rwernis@berkeley.edu)

**Abstract.** Gas- and particle–phase molecular markers provide highly specific information about the sources and atmospheric
processes that contribute to air pollution. In urban areas, major sources of pollution are changing as regulation selectively
mitigates some pollution sources and climate change impacts the surrounding environment. In this study, a Comprehensive
Thermal Desorption Aerosol Gas Chromatograph (cTAG) was used to measure volatile, intermediate volatility, and semi-
volatile molecular markers every other hour over a 10-day period from 11 April to 21 April 2018 in suburban Livermore,
California. Source apportionment via Positive Matrix Factorization (PMF) was performed to identify major sources of
pollution. The PMF analysis identified 13 components, including emissions from gasoline, consumer products, biomass
burning, secondary oxidation, aged regional transport, and several factors associated with single compounds or specific
events with unique compositions. The gasoline factor had a distinct morning peak in concentration but lacked a
corresponding evening peak, suggesting commute-related traffic emissions are dominated by cold starts in residential areas.
More monoterpene and monoterpenoid mass was assigned to consumer product emissions than biogenic sources,
underscoring the increasing importance of volatile chemical products to urban emissions. Daytime isoprene concentrations
were controlled by biogenic sunlight- and temperature-dependent processes, mediated by strong midday mixing, but gasoline
was found to be the dominant and likely only source of isoprene at night. Biomass burning markers indicated residential
wood burning activity remained an important pollution source even in the springtime. This study demonstrates the utility of
specific high-time-resolution molecular marker measurements across a wide range of volatility in more comprehensively
describing pollution source profiles than a narrower volatility range would allow.

## 1 Introduction

Organic carbon in the atmosphere spans more than 15 orders of magnitude of volatility (Jimenez et al., 2009;
Donahue et al., 2011). Some of the organic carbon is emitted directly as primary organic aerosol (POA), but most organic
carbon is emitted in the gas phase as thousands of distinct compounds (Goldstein and Galbally, 2007). The most volatile
class, volatile organic compounds (VOCs), exist exclusively in the gas phase. Many VOCs are toxic or contribute to





respiratory illness (Srivastava et al., 2005; Nurmatov et al., 2013). They play a critical role in urban and regional ozone formation (National Research Council, 1992; Atkinson, 2000) and produce lower vapor pressure compounds via atmospheric oxidation reactions (Seinfeld and Pankow, 2003) which form secondary organic aerosol (SOA). Intermediate volatility organic compounds (IVOCs, defined as having an effective saturation concentration $C^*$ of $10^3$ to $10^6$ µg m$^{-3}$) and semi-

volatile organic compounds (SVOCs, $C^*$ of $10^{-1}$ to $10^3$ µg m$^{-3}$), can partition between gas and particle phases in the atmosphere and can account for a large fraction of total organic aerosol (OA) (Robinson et al., 2007; Weitkamp et al., 2007; Chan et al., 2009; de Gouw et al., 2011; Lim and Ziemann, 2009; Presto et al., 2010). OA is a major source of uncertainty for radiative forcing predictions (Myhre et al., 2013) and negatively impacts human health (Brunekreef and Holgate, 2002; Nel, 2005; Lippmann and Chen, 2009). In the United States, it comprises 30-80 % of annually averaged particulate matter with a

diameter of less than 2.5 µm (PM$_{2.5}$; Hand et al., 2013). PM$_{2.5}$ is regulated by the U.S. Environmental Protection Agency due to its adverse impacts on human health (Ridley et al., 2018).

Sources of VOCs, IVOCs and SVOCs in urban areas are changing. Historically, vehicle exhaust has been responsible for the bulk of VOC emissions in polluted urban areas such as Los Angeles, but VOC emissions from gasoline vehicles have decreased by almost two orders of magnitude between 1960 and 2010 (Warneke et al., 2012). Diesel emissions

have also decreased, though to a lesser extent (McDonald et al., 2013). As a result, sources of non-motor-vehicle organic carbon in urban areas have increased in relative importance. McDonald et al. (2018) recently showed that volatile chemical products, including fragrances, solvents, pesticides, coatings, inks, adhesives and cleaning agents, were responsible for more VOC emissions by mass than vehicle and upstream petrochemical emissions combined in Los Angeles. Coggon et al. (2021) demonstrated the same for New York City and showed that monoterpene emissions from fragrances in Manhattan rivaled

those of a comparably sized U.S. forest. Furthermore, global climate change directly affects biogenic emissions of reactive organic carbon (Heald et al., 2008; Lin et al., 2016) and, in the American West, has led to increased emissions from forest fires (Hurteau et al., 2014; Westerling, 2016); indirectly, climate change may affect pollution patterns related to heating and cooling indoor spaces. Given the changing urban and regional environments, sources of pollution in urban and suburban areas need to be continually reevaluated to improve predictions of ozone and SOA formation and inform policy decision

making around emission reductions and mitigation of pollution impacts.

Many organic compounds have more than one source category contributing to their abundance in the atmosphere. Source apportionment models may be used to identify the underlying sources behind the measured concentrations of speciated organics based on the variation of individual concentration timelines. Positive Matrix Factorization (PMF; Paatero and Tapper, 1994; Hopke, 2016) is a technique often used to apportion VOCs (e.g. Brown et al., 2007; Yuan et al., 2009;

Yuan et al., 2012) and compositionally resolved particulate matter (e.g. Wang et al., 2019a; Li et al., 2020) into different factors contributing to their measured abundance. PMF does not require pollution source profiles as inputs, making it an attractive approach to analyzing contributions to ambient atmospheric abundances when not all possible sources are known.

In this work PMF was applied to concentration timelines of a suite of VOCs, IVOCs and SVOCs in the gas and particle phases measured every other hour by the Comprehensive Thermal Desorption Aerosol Gas Chromatograph (cTAG)





over a 10-day period in Livermore, CA. Our aim was to identify and understand the major sources of pollution in a suburban setting in the context of changing emission controls and dominant sources observed in other urban and suburban areas nationwide. With compounds encompassing such a wide range in volatility and degree of oxidation, we sought to more comprehensively describe the composition of the identified pollution sources. We examine the detailed temporal patterns of the different factors and provide possible explanations for their variability based on likely activity patterns, atmospheric chemistry and the meteorology of the region.

## 2 Methods

### 2.1 Sampling site

Livermore, California is a suburban city located on the eastern edge of the San Francisco Bay Area in the Livermore Valley. It is subject to prevailing winds from the larger Bay Area region to the west, bringing primary pollutants that, combined with optimal conditions in the city itself for photochemical smog formation, often lead to the highest ozone levels in the Bay Area (Flagg et al., 2020). Regional transport is also possible from the neighboring San Joaquin Valley to the east. In wintertime, temperature inversions and low wind speeds can lead to elevated particulate matter concentrations.

Speciated VOC, IVOC and SVOC measurements were collected at the May Nissen Swim Center, 685 Rincon Avenue in Livermore (37.687°N, 121.784°W). The Swim Center is approximately 60 m west of Rincon Avenue, the closest road, and 1.4 km south of Interstate 580. An uncovered outdoor swimming pool, closed to public access for the season, was 10 m to the northeast of the sampling site. Approximately 100 m to the north at 793 Rincon Ave, the Livermore Bay Area Air Quality Management District Monitoring Site obtained hourly measurements of ozone, temperature, black carbon and wind direction and speed used in this analysis.

### 2.2 cTAG measurements of VOCs, IVOCs and SVOCs

Hourly VOC, IVOC and SVOC speciated measurements were collected and analyzed by cTAG between 9 April and 11 May 2018. This work analyzes data from a 10-day focus period between 11 April and 21 April when the cTAG instrument was operating optimally. Ambient air from approximately 5 m above ground was pulled through a 25 cm diameter duct to the inlet of cTAG at 1000 L min$^{-1}$. This high flow rate ensured small residence times for semivolatile analytes of interest in the ducting and thus negligible partitioning to the walls of the duct. cTAG is described elsewhere in detail (Wernis et al., 2021). Briefly, 10.1 L min$^{-1}$ of ambient air is pulled from the duct through a cyclone (PM$_{2.5}$ cut point) and split. To measure gas-particle partitioning, on every other sample 10.0 L min$^{-1}$ is passed through a denuder to remove gas phase compounds, resulting in alternating hourly measurements of total gas plus particle concentration versus particle phase only concentration. This 10.0 L min$^{-1}$ is then pulled through a coated metal mesh filter cell held at 30 °C which collects IVOCs and SVOCs between C$_{14}$ and C$_{32}$ alkane equivalent volatility. The remaining 100 sccm is pulled through a bed of adsorbent materials also held at 30 °C designed to efficiently collect VOCs and IVOCs between C$_5$ and C$_{16}$ alkane



equivalent volatility. After the 22 minute sampling period, the collected samples are analyzed in series. The I/VOC collector is desorbed in helium onto the I/VOC channel GC column, which is designed for separation of volatile organics (Restek metal MXT-624, 30 m, 0.32 mm inner diameter, 1.8 μm phase). Column effluent enters the high-resolution time-of-flight mass spectrometer (HRToFMS, TOFWERK), operated at 70 eV electron impact ionization energy to generate individual

mass spectra for the separated compounds. GC analysis time is 25 min, with an initial hold for 1 min at 40 °C followed by a 10 °C min$^{-1}$ ramp to 250 °C and a 3 min hold at 250 °C.

During GC and HRToFMS analysis of the I/VOC sample, desorption begins for the SVOC sample in helium saturated with the derivatization agent *N*-methyl-*N*-(trimethylsilyl) trifluoroacetamide (MSTFA). The MSTFA reacts with OH groups on polar SVOCs to make them less polar and thus more amenable to measurement on the SVOC channel GC

column, which is optimized for nonpolar organics. After a reconcentration step which removes excess MSTFA and allows for faster transfer of low volatility species from the collection filter cell, the sample is transferred to the GC column (Restek metal MXT-5, 20 m, 0.18 mm inner diameter, 2 μm phase) for separation and detection by the HRToFMS. GC analysis time is 19 min, with an initial hold for 1 min at 50 °C followed by a 20 °C min$^{-1}$ ramp to 330 °C and a 4 min hold at 330 °C.

### 2.2.1 Compound identification

Two chromatograms with mass spectral information are generated every hour – one for I/VOCs and one for SVOCs. Typical total ion chromatograms (TICs) can contain hundreds to thousands of compounds, often leading to overlapping peaks. Single ion chromatograms (SICs) have far fewer overlaps; thus integrated peaks on the SIC corresponding to a prominent mass-to-charge ratio in the target compound's mass spectrum are used as the basis for quantification. Background subtraction isolates the mass spectral fingerprint of the target compound for better matching with

searches against the 2020 National Institute of Standards and Technology (NIST) mass spectral library (NIST Standard Reference Database 1A, 2022) and authentic standards analyzed on cTAG or similar instruments. The retention index (RI) describes the relative location of a compound in a chromatogram to a series of reference compounds. The *n*-alkane retention index for a compound *i* is defined as:

$$\text{RI} = 100 \times \left[ n + \frac{t_i - t_n}{t_{n+1} - t_n} \right] \tag{1}$$

where *n* is the number of carbon atoms of the *n*-alkane that elutes immediately prior to compound *i* and *t* is the retention

time. *N*-alkane RI comparisons between the target compound and candidate matches aid in conclusively identifying the target compound.

### 2.2.2 Compound quantification

The integrated peak area calculated for a given compound on a given chromatogram depends on a number of variable factors including but not limited to drift in the response of the detector, ion source cleanliness, transfer efficiency

between the collectors and the GC columns and derivatization efficiency (SVOC channel only). Furthermore, these variables





can affect the integrated peak area in a compound-dependent way. To account for these variables, compound quantification involves a multistep process. On the SVOC channel:

(1) A constant quantity of a suite of isotopically labeled compounds with a variety of volatilities and functional groups comprising an internal standard mixture was automatically injected (Isaacman et al., 2011) onto the filter cell after every sample collection and before thermal desorption. These compounds were analyzed along with the ambient sample. Variations in integrated peak area for these compounds capture the instrument response variability described above, improving measurement precision. Ambient compound peak areas are normalized by the SIC peak area of the internal standard that most closely matches it in volatility and functionality.

(2) In the laboratory after the field campaign, an external standard mixture consisting of 218 compounds was injected onto the filter cell in varying concentrations to obtain a 6-point calibration curve for those compounds. The same internal standard mixture used during ambient sampling was injected during calibration runs, and peak-integrated ion signals of the calibrant compounds were normalized by the most suitable internal standard.

(3) External standard calibration curves are fit with a least-squares regression. Ambient compounds with exact matches in the external standard mixture have the slope $b$ and y-intercept $a$ from this fit applied directly to convert peak area $S_A$ to mass $M_A$. For ambient compounds without exact external standard matches the final mass $M_A$ is adjusted by the ratio of the fraction of signal represented by each SIC in each TIC:

$$M_A = \left(\frac{S_A - a}{b}\right)\left(\frac{f_C}{f_A}\right) \tag{2}$$

$$f = s_{\text{SIC}} / \sum_{i=4}^{400} s_i \tag{3}$$

where $f$ is the fraction of TIC signal represented by the signal at the SIC used for quantification ($f_A$ is for the analyte and $f_C$ is for the external standard calibration compound), $s_{\text{SIC}}$ is the quantification SIC signal, and the denominator is the sum of all SIC signals (i.e. the TIC signal). $i$ is a mass to charge ratio, which ranges from 4 to 400 for this campaign and analysis.

I/VOC channel calibration is functionally equivalent to SVOC channel calibration with a couple important differences. (1) A single gas-phase internal standard, neohexane, was introduced at the inlet at a constant 100 parts-per-trillion throughout every sampling period for run-to-run normalization. All analytes are normalized by neohexane. (2) External standard compounds originated from a Photochemical Assessment Monitoring Stations 57-component commercial standard gas cylinder (Scott-Specialty), a gas cylinder with a custom mixture (Apel-Riemer Environmental, Inc., 2019), and two custom liquid mixtures. Standards were delivered with the dynamic dilution system developed for cTAG (Wernis et al., 2021), generating 6-point calibrations. Neohexane could not be sampled during calibrations, so was sampled before and after each set of calibration runs and the average was used to normalize all calibration points.



### 2.2.3 Uncertainty in reported concentration

There are two distinct types of uncertainty affecting reported concentrations for compounds measured by cTAG. The first, uncertainty around accuracy, arises from calibration, which affects all ambient data points of a given compound identically. Uncertainty on the least squares fit of the calibration data and uncertainty arising from the lack of an authentic external standard are uncertainties of this type. The second, uncertainty of precision, arises from run-to-run variability (Sect. 2.2.2), which we assume to be independent between samples. Internal standard normalization greatly mitigates this source of

uncertainty but does not eliminate it. The uncertainty that remains depends on the choice of internal standard. The total uncertainty in the reported concentration of a given compound is all of these sources of uncertainty added together in quadrature.

#### 2.2.3.1 Accuracy uncertainty

       Accuracy uncertainty from the least squares fit is generally limited to the uncertainty of the slope, $\Delta_b$, as the y-

intercept is kept fixed at 0 for compounds without background contamination, which is the great majority of them. The percent uncertainty from calibration fit $UC_b$ is thus:

$$UC_b = 100 \times \left(\frac{\Delta_b}{b}\right) \tag{4}$$

In practice, this adds less than 5 % uncertainty to the total for a given compound.

       Compounds without an authentic external standard have an additional source of accuracy uncertainty arising from the use of a surrogate standard. This source is much greater than that from the calibration fit, and impossible to quantify

individually for each compound in the absence of an authentic standard to serve as the control. Jaoui et al. (2005) report approximately 30 % error from this step; we conservatively estimate 50 % uncertainty for surrogate standard use.

#### 2.2.3.2 Precision uncertainty

       Precision uncertainty is based on choice of internal standard. To estimate this source of uncertainty, all possible pairs of internal standards were ratioed for all ambient data points and each distribution of ratios analyzed, a technique used

with previous TAG instruments (Isaacman et al., 2014). Ideally, all internal standards would vary proportionally and the ratio between two standards would remain constant, implying no error from internal standard choice. In practice the ratio varies between samples and the relative standard deviation (RSD, standard deviation divided by mean) of the ratios provides an estimate of the precision uncertainty.

       Figure S1 shows the relative standard deviations of the internal standard ratios for all internal standards used for

normalization in this analysis and Fig. S2 shows examples of the distributions of ratios. Overall, hydrocarbons paired with hydrocarbons have the lowest RSDs, especially for compounds with similar RIs. An exception is made for compounds with a RI below 1400, where occasional losses during the refocusing step due to high ambient temperature increase the variability. Ambient compounds in this category are normalized by deuterated n-tetradecane. Oxygenated internal standards exhibit the greatest RSD values whether they are paired with hydrocarbons or other oxygenates. Ambient oxygenated compounds are

thus assigned the highest uncertainty and are normalized by the nearest deuterated oxygenate if their RIs are within 200 and




the nearest hydrocarbon otherwise. Table S1 summarizes the categories of precision uncertainty assigned to ambient compounds for this analysis.

**2.3 Positive matrix factorization**

Positive Matrix Factorization is a mathematical source apportionment technique that groups measured ambient
compounds based on their covariance in time, taking into account their measurement uncertainty (Paatero and Tapper, 1994). PMF is a receptor-only model that requires no a priori information about pollution sources, instead inferring source composition from the compound groupings in the solution. The solution is constrained to non-negative values and assumes source profiles do not vary with time. It takes the form of three matrices **G**, **F** and **E** such that

$$x_{ij} = \sum_{k=1}^{p} g_{ik} f_{kj} + e_{ij} \qquad (5)$$

where $x_{ij}$ is an element of the $m$ x $n$ matrix **X** of input data. As applied to this cTAG dataset, the $m$ rows of **X** are the
individual compounds and the $n$ columns of **X** are the sample times. Element $g_{ik}$ of **G** represents the source contribution of the $i$th compound to the $k$th factor, and element $f_{kj}$ of **F** represents the $k$th factor at sample $j$. **E** is the $m$ x $n$ matrix of residual values. Crucially, the total number of factors $p$ is an input to the model, requiring the model user to use PMF solution diagnostics and outside information to determine the most meaningful number of factors. The solution is determined by minimizing the sum of squares of error-weighted residuals, known as the quality of fit parameter $Q$, given by

$$Q = \sum_{i=1}^{m} \sum_{j=1}^{n} \left( \frac{e_{ij}}{\sigma_{ij}} \right)^2 \qquad (6)$$

where $\sigma_{ij}$ is the uncertainty in concentration units for compound $i$ at sample $j$ and $e_{ij}$ is the corresponding element of matrix **E**.

This analysis is performed on 2 hour time resolution data aligning with the gas plus particle phase concentration measurements on the SVOC channel. For the PMF, I/VOC channel data that coincides with particle only measurements on the SVOC channel are not used, though the full hourly resolution I/VOC data are used for other data analysis (e.g.,
correlations) and in figures. Similarly, particle only data on the SVOC channel are not used in the PMF analysis but partitioning information does inform some of the interpretation of factor results.

In this analysis the precision uncertainty (Sect. 2.2.3.2) is the only uncertainty assigned to each compound for use in the PMF model, since the model assumes uncertainty between samples is independent, which is not true of the accuracy uncertainties described above (Sect. 2.2.3.1). In the final PMF solution, accuracy uncertainty increases the uncertainty of the
factor timelines (while preserving ratios between individual data points) but not the source contributions. PMF modelling was carried out with the U.S. Environmental Protection Agency PMF 5.0 program (Norris et al., 2014). The program automatically handles uncertainty for concentrations near the detection limit by applying a smooth function to the uncertainty between the input percent uncertainty (applies to concentrations ≫ the detection limit) and a large fixed fraction



of the detection limit (applies to concentrations at or below the detection limit). We use the "robust" mode of the PMF

algorithm, which limits the weight of outliers ($|e_{ij}/\sigma_{ij}| > 4$) by increasing the uncertainty of those outliers. We also explore the stability of the most plausible solutions by varying FPEAK, a parameter which applies rotations by adding **G** columns to each other and subtracting **F** rows from each other or vice versa (Paatero and Hopke, 2009).

## 3 Results and discussion

Of 163 ambient compounds processed in the dataset, 123 were used in the PMF analysis. The remaining 40 were

excluded for one of the following reasons: the compound's concentration was below the detection limit more than 90 % of the time (27 compounds), the compound was determined to have too much instrument contamination to be quantifiable (7 compounds), the compound could not be definitively identified (3 compounds), the compound's transfer losses were not well characterized due to being outside cTAG's designed optimal volatility range (2 compounds), or the compound is nonreactive in the atmosphere (1 compound).

Of the 123 compounds used in the PMF analysis, 58 compounds were measured on the I/VOC channel and the remaining 65 were measured on the SVOC channel. Table 1 shows the compounds included in the PMF analysis. Major compound categories represented include branched and linear alkanes and aromatics important for photochemical smog formation, monoterpenes and other biogenic compounds, polycyclic aromatic hydrocarbons, biomass burning markers, alkanoic acids, chlorobenzenes, and plasticizers and other industrial chemicals.

## 3.1 Positive matrix factorization solution

We find the 13 factor PMF solution to best resolve the pollution sources in Livermore in Spring 2018. Solutions between 3 and 16 factors were considered and are contrasted in Sect. S2. In summary, solutions with additional factors have less uniqueness between factors and the source profiles of the additional factors are either not physically meaningful or are too similar to factors present in the 13 factor solution. Solutions with fewer factors fail to separate factors with meaningful physical interpretations and do not incorporate one of the largest reductions in $Q/Q_{exp}$.







**Table 1. Compounds measured by cTAG and included in the PMF analysis. The compound index is used in Fig. 2. PAH = Polycyclic Aromatic Hydrocarbons; Ox. = Oxidation; Alk. Acids = Alkanoic Acids; HCs = Hydrocarbons; BB = Biomass Burning; THM = Trihalomethanes; DCB = Dichlorobenzenes; DEHA = Bis(2-ethylhexyl) adipate; DEHP = Bis(2-ethylhexyl) phthalate; D4 = Octamethylcyclotetrasiloxane; D5 = Decamethylcyclopentasiloxane; PCBTF = Parachlorobenzotrifluoride.**

| Compound Class | Compound Index and Name | CAS # | Meas. Channel | Mean ± S. Deviation (ng m$^{-3}$) | Mean ± S. Deviation (ppt) | Highest Mass Fraction & Factor | |
|---|---|---|---|---|---|---|---|
| Saturated HCs | (1) Methylcyclopentane | 96-37-7 | I/VOC | 290 ± 320 | 85 ± 93 | 0.33 | 3 |
| Saturated HCs | (2) 2-Methylhexane | 591-76-4 | I/VOC | 140 ± 170 | 33 ± 40 | 0.45 | 3 |
| Saturated HCs | (3) 3-Methylhexane | 589-34-4 | I/VOC | 360 ± 190 | 87 ± 47 | 0.19 | 1 |
| Saturated HCs | (4) 2,3-Dimethylpentane | 565-59-3 | I/VOC | 130 ± 180 | 33 ± 44 | 0.31 | 3 |
| Saturated HCs | (5) Cyclohexane | 110-82-7 | I/VOC | 120 ± 140 | 35 ± 42 | 0.44 | 3 |
| Saturated HCs | (6) 2,2,4-Trimethylpentane | 540-84-1 | I/VOC | 400 ± 503 | 86 ± 108 | 0.29 | 3 |
| Saturated HCs | (7) Benzene | 71-43-2 | I/VOC | 610 ± 310 | 190 ± 100 | 0.23 | 3 |
| Saturated HCs | (8) Heptane (C$_7$) | 142-82-5 | I/VOC | 170 ± 180 | 42 ± 43 | 0.31 | 3 |
| Saturated HCs | (9) Methylcyclohexane | 108-87-2 | I/VOC | 120 ± 140 | 30 ± 34 | 0.35 | 3 |
| Saturated HCs | (10) 2,3,4-Trimethylpentane | 565-75-3 | I/VOC | 140 ± 190 | 29 ± 40 | 0.29 | 3 |
| Saturated HCs | (11) 2-Methylheptane | 592-27-8 | I/VOC | 370 ± 470 | 79 ± 102 | 0.32 | 3 |
| Saturated HCs | (12) 3-Methylheptane | 589-81-1 | I/VOC | 84 ± 100 | 18 ± 21 | 0.36 | 3 |
| Saturated HCs | (13) Octane (C$_8$) | 111-65-9 | I/VOC | 86 ± 88 | 18 ± 19 | 0.26 | 3 |
| Saturated HCs | (14) Toluene | 108-88-3 | I/VOC | 930 ± 960 | 250 ± 250 | 0.30 | 3 |
| Saturated HCs | (15) Nonane (C$_9$) | 111-84-2 | I/VOC | 64 ± 68 | 12 ± 13 | 0.20 | 3 |
| Saturated HCs | (16) Ethylbenzene | 100-41-4 | I/VOC | 190 ± 220 | 43 ± 51 | 0.36 | 3 |
| Saturated HCs | (17) m-Xylene + p-Xylene | 108-38-3, 106-42-3 | I/VOC | 670 ± 840 | 150 ± 190 | 0.36 | 3 |
| Saturated HCs | (18) o-xylene | 95-47-6 | I/VOC | 290 ± 360 | 66 ± 83 | 0.37 | 3 |
| Saturated HCs | (19) Styrene | 100-42-5 | I/VOC | 52 ± 61 | 12 ± 14 | 0.36 | 3 |
| Saturated HCs | (20) Cumene | 98-82-8 | I/VOC | 10 ± 12 | 2.1 ± 2.4 | 0.23 | 3 |
| Saturated HCs | (21) n-Propylbenzene | 103-65-1 | I/VOC | 39 ± 47 | 8.0 ± 9.6 | 0.36 | 3 |
| Saturated HCs | (22) Decane (C$_{10}$) | 124-18-5 | I/VOC | 63 ± 72 | 11 ± 12 | 0.18 | 5 |
| Saturated HCs | (23) m-Ethyltoluene | 620-14-4 | I/VOC | 110 ± 150 | 23 ± 31 | 0.41 | 3 |
| Saturated HCs | (24) 1,3,5-Trimethylbenzene | 108-67-8 | I/VOC | 68 ± 99 | 14 ± 20 | 0.44 | 3 |
| Saturated HCs | (25) o-Ethyltoluene | 611-14-3 | I/VOC | 55 ± 73 | 11 ± 15 | 0.41 | 3 |
| Saturated HCs | (26) 1,2,4-Trimethylbenzene | 95-63-6 | I/VOC | 290 ± 410 | 60 ± 84 | 0.41 | 3 |
| Saturated HCs | (27) 1,2,3-Trimethylbenzene | 526-73-8 | I/VOC | 55 ± 78 | 11 ± 16 | 0.42 | 3 |





| Compound Class | Compound Index and Name | CAS # | Meas. Channel | Mean ± S. Deviation (ng m$^{-3}$) | Mean ± S. Deviation (ppt) | Highest Mass Fraction & Factor | |
|---|---|---|---|---|---|---|---|
| Saturated HCs | (28) m-Diethylbenzene | 141-93-5 | I/VOC | 9.4 ± 14.3 | 1.7 ± 2.6 | 0.43 | 3 |
| Saturated HCs | (29) p-Diethylbenzene | 105-05-5 | I/VOC | 63 ± 98 | 12 ± 18 | 0.41 | 3 |
| Saturated HCs | (30) Undecane (C$_{11}$) | 1120-21-4 | I/VOC | 53 ± 92 | 8.3 ± 14.4 | 0.18 | 5 |
| Saturated HCs | (31) Dodecane (C$_{12}$) | 112-40-3 | I/VOC | 31 ± 67 | 4.5 ± 9.6 | 0.22 | 11 |
| Saturated HCs | (32) Tridecane (C$_{13}$) | 629-50-5 | I/VOC | 56 ± 123 | 7.4 ± 16.4 | 0.25 | 11 |
| Saturated HCs | (33) Tetradecane (C$_{14}$) | 629-59-4 | I/VOC | 49 ± 71 | 6.1 ± 8.8 | 0.32 | 5 |
| Saturated HCs | (34) Pentadecane (C$_{15}$) | 629-62-9 | SVOC | 15 ± 11 | 1.7 ± 1.3 | 0.26 | 5 |
| Saturated HCs | (35) Hexadecane (C$_{16}$) | 544-76-3 | SVOC | 10 ± 10 | 1.1 ± 1.1 | 0.22 | 11 |
| Saturated HCs | (36) Heptadecane (C$_{17}$) | 629-78-7 | SVOC | 11 ± 10 | 1.1 ± 1.0 | 0.21 | 6 |
| Saturated HCs | (37) Pristane | 1921-70-6 | SVOC | 5.3 ± 4.7 | 0.48 ± 0.43 | 0.24 | 11 |
| Saturated HCs | (38) Octadecane (C$_{18}$) | 593-45-3 | SVOC | 6.8 ± 6.1 | 0.66 ± 0.59 | 0.26 | 6 |
| Saturated HCs | (39) Phytane | 638-36-8 | SVOC | 2.8 ± 2.3 | 0.24 ± 0.20 | 0.24 | 6 |
| Saturated HCs | (40) Nonadecane (C$_{19}$) | 629-92-5 | SVOC | 3.90 ± 2.2 | 0.36 ± 0.20 | 0.22 | 10 |
| Saturated HCs | (41) Eicosane (C$_{20}$) | 112-95-8 | SVOC | 4.0 ± 1.9 | 0.34 ± 0.17 | 0.26 | 4 |
| Saturated HCs | (42) Heneicosane (C$_{21}$) | 629-94-7 | SVOC | 8.50 ± 4.9 | 0.70 ± 0.41 | 0.31 | 4 |
| Saturated HCs | (43) Docosane (C$_{22}$) | 629-97-0 | SVOC | 9.0 ± 6.5 | 0.71 ± 0.51 | 0.37 | 2 |
| Saturated HCs | (44) Tricosane (C$_{23}$) | 638-67-5 | SVOC | 6.3 ± 5.40 | 0.47 ± 0.41 | 0.49 | 2 |
| Saturated HCs | (45) Tetracosane (C$_{24}$) | 646-31-1 | SVOC | 2.3 ± 2.0 | 0.16 ± 0.14 | 0.54 | 4 |
| Saturated HCs | (46) Pentacosane (C$_{25}$) | 629-99-2 | SVOC | 4.0 ± 2.8 | 0.28 ± 0.19 | 0.43 | 4 |
| Saturated HCs | (47) Hexacosane (C$_{26}$) | 630-01-3 | SVOC | 2.1 ± 1.7 | 0.14 ± 0.11 | 0.50 | 4 |
| Saturated HCs | (48) Heptacosane (C$_{27}$) | 593-49-7 | SVOC | 1.5 ± 1.2 | 0.09 ± 0.08 | 0.31 | 4 |
| Alk. Acids | (49) Octanoic (C$_8$) Acid | 124-07-2 | SVOC | 11 ± 5 | 1.9 ± 0.8 | 0.20 | 4 |
| Alk. Acids | (50) Nonanoic (C$_9$) Acid | 112-05-0 | SVOC | 41 ± 16 | 6.3 ± 2.5 | 0.22 | 4 |
| Alk. Acids | (51) Decanoic (C$_{10}$) Acid | 334-48-5 | SVOC | 17 ± 7 | 2.4 ± 1.0 | 0.24 | 4 |
| Alk. Acids | (52) Undecanoic (C$_{11}$) Acid | 112-37-8 | SVOC | 7.8 ± 3.7 | 1.0 ± 0.5 | 0.32 | 4 |
| Alk. Acids | (53) Dodecanoic (C$_{12}$) Acid | 143-07-7 | SVOC | 12 ± 6 | 1.5 ± 0.7 | 0.29 | 4 |
| Alk. Acids | (54) Tridecanoic (C$_{13}$) Acid | 638-53-9 | SVOC | 5.8 ± 3.3 | 0.66 ± 0.38 | 0.40 | 4 |
| Alk. Acids | (55) Tetradecanoic (C$_{14}$) Acid | 544-63-8 | SVOC | 15 ± 8 | 1.6 ± 0.9 | 0.31 | 4 |
| Alk. Acids | (56) Pentadecanoic (C$_{15}$) Acid | 1002-84-2 | SVOC | 9.5 ± 6.7 | 0.96 ± 0.68 | 0.46 | 4 |




| Compound Class | Compound Index and Name | CAS # | Meas. Channel | Mean ± S. Deviation (ng m$^{-3}$) | Mean ± S. Deviation (ppt) | Highest Mass Fraction & Factor | |
|---|---|---|---|---|---|---|---|
| Alk. Acids | (57) Palmitic (C$_{16}$) Acid | 57-10-3 | SVOC | 66 ± 63 | 6.4 ± 6.1 | 0.26 | 4 |
| Alk. Acids | (58) Heptadecanoic (C$_{17}$) Acid | 506-12-7 | SVOC | 24 ± 27 | 2.1 ± 2.5 | 0.42 | 4 |
| Alk. Acids | (59) Stearic (C$_{18}$) Acid | 57-11-4 | SVOC | 1100 ± 1300 | 93 ± 116 | 0.31 | 4 |
| Alk. Acids | (60) Azelaic (C$_9$) Acid | 123-99-9 | SVOC | 96 ± 115 | 12 ± 15 | 0.33 | 4 |
| Other | (61) C$_{16}$ acid methyl ester | 112-39-0 | SVOC | 20 ± 14 | 1.8 ± 1.3 | 0.20 | 10 |
| Other | (62) Nonanal | 124-19-6 | I/VOC | 890 ± 1260 | 150 ± 220 | 0.25 | 11 |
| Terpenoid | (63) Isoprene | 78-79-5 | I/VOC | 190 ± 220 | 68 ± 79 | 0.73 | 12 |
| Terpenoid | (64) Camphene | 79-92-5 | I/VOC | 19 ± 22 | 3.5 ± 3.9 | 0.29 | 6 |
| Terpenoid | (65) Camphor | 76-22-2 | I/VOC | 74 ± 60 | 12 ± 10 | 0.29 | 8 |
| Terpenoid | (66) alpha-Pinene | 80-56-8 | I/VOC | 480 ± 1370 | 87 ± 246 | 0.35 | 5 |
| Terpenoid | (67) beta-Pinene | 127-91-3 | I/VOC | 49 ± 54 | 8.8 ± 9.7 | 0.25 | 6 |
| Terpenoid | (68) Limonene | 138-86-3 | I/VOC | 160 ± 300 | 28 ± 53 | 0.66 | 5 |
| Terpenoid | (69) 3-Carene | 13466-78-9 | I/VOC | 18 ± 25 | 3.3 ± 4.4 | 0.35 | 5 |
| Terpenoid | (70) Eucalyptol | 470-82-6 | I/VOC | 310 ± 250 | 49 ± 40 | 0.28 | 8 |
| Terpenoid | (71) Aromadendrene | 109119-91-7 | SVOC | 0.9 ± 0.6 | 0.10 ± 0.08 | 0.37 | 4 |
| Terpene Ox. | (72) Pinic Acid | 473-73-4 | SVOC | 2.5 ± 1.7 | 0.32 ± 0.22 | 0.37 | 8 |
| Terpene Ox. | (73) Pinonic Acid | 473-72-3 | SVOC | 21 ± 15 | 2.8 ± 1.9 | 0.52 | 8 |
| Other | (74) Methyl Salicylate | 119-36-8 | SVOC | 3.3 ± 1.8 | 0.52 ± 0.29 | 0.22 | 8 |
| Other | (75) alpha-Isomethyl ionone | 127-51-5 | SVOC | 0.70 ± 0.99 | 0.08 ± 0.12 | 0.45 | 5 |
| Other | (76) Dibenzofuran | 132-64-9 | SVOC | 1.6 ± 0.9 | 0.23 ± 0.13 | 0.21 | 8 |
| Other | (77) Benzophenone | 119-61-9 | SVOC | 2.9 ± 1.8 | 0.38 ± 0.24 | 0.24 | 8 |
| Other | (78) Phthalic Anhydride | 85-44-9 | SVOC | 0.72 ± 0.53 | 0.12 ± 0.09 | 0.42 | 4 |
| Plasticizer | (79) Dibutyl Phthalate | 84-74-2 | SVOC | 1.2 ± 0.8 | 0.11 ± 0.07 | 0.26 | 4 |
| Plasticizer | (80) Diethyl Phthalate | 84-66-2 | SVOC | 3.7 ± 2.0 | 0.41 ± 0.22 | 0.20 | 10 |
| Plasticizer | (81) Dimethyl Phthalate | 131-11-3 | SVOC | 0.61 ± 0.31 | 0.08 ± 0.04 | 0.23 | 8 |
| Plasticizer | (82) Benzyl butyl phthalate | 85-68-7 | SVOC | 2.6 ± 1.5 | 0.20 ± 0.12 | 0.28 | 10 |
| Plasticizer | (83) DEHA | 103-23-1 | SVOC | 500 ± 600 | 33 ± 40 | 0.62 | 4 |
| Plasticizer | (84) DEHP | 117-81-7 | SVOC | 10 ± 10 | 0.7 ± 0.6 | 0.46 | 4 |
| BB | (85) Furfural | 98-01-1 | I/VOC | 160 ± 130 | 41 ± 33 | 0.29 | 9 |
| BB | (86) Levoglucosan | 498-07-7 | SVOC | 18 ± 27 | 2.7 ± 4.1 | 0.67 | 9 |
| BB | (87) Galactosan | 644-76-8 | SVOC | 0.6 ± 1.1 | 0.09 ± 0.17 | 0.68 | 9 |
| BB | (88) Mannosan | 14168-65-1 | SVOC | 5.3 ± 8.5 | 0.81 ± 1.3 | 0.62 | 9 |
| BB | (89) Catechol | 120-80-9 | SVOC | 0.7 ± 0.3 | 0.16 ± 0.07 | 0.22 | 4 |





| Compound Class | Compound Index and Name | CAS # | Meas. Channel | Mean ± S. Deviation (ng m$^{-3}$) | Mean ± S. Deviation (ppt) | Highest Mass Fraction & Factor | |
|---|---|---|---|---|---|---|---|
| BB | (90) p-Anisic Acid | 100-09-4 | SVOC | 2.5 ± 1.2 | 0.41 ± 0.20 | 0.26 | 10 |
| BB | (91) 4-Hydroxybenzoic Acid | 99-96-7 | SVOC | 3.1 ± 1.2 | 0.56 ± 0.21 | 0.35 | 1 |
| BB | (92) Vanillin | 121-33-5 | SVOC | 8.2 ± 6.7 | 1.3 ± 1.1 | 0.21 | 9 |
| BB | (93) Vanillic Acid | 121-34-6 | SVOC | 1.0 ± 1.0 | 0.14 ± 0.14 | 0.46 | 9 |
| BB | (94) Syringaldehyde | 134-96-3 | SVOC | 0.9 ± 1.8 | 0.13 ± 0.24 | 0.66 | 9 |
| BB | (95) Syringic Acid | 530-57-4 | SVOC | 0.8 ± 1.0 | 0.10 ± 0.13 | 0.59 | 9 |
| BB | (96) 4-Nitrocatechol | 3316-09-4 | SVOC | 1.7 ± 2.3 | 0.27 ± 0.37 | 0.41 | 9 |
| Other | (97) 2-Nitrophenol | 88-75-5 | SVOC | 1.3 ± 1.2 | 0.23 ± 0.22 | 0.25 | 8 |
| Other | (98) 4-Nitrophenol | 100-02-7 | SVOC | 3.1 ± 2.6 | 0.54 ± 0.45 | 0.38 | 4 |
| Other | (99) Palmitoleic Acid | 373-49-9 | SVOC | 2.8 ± 10.6 | 0.24 ± 0.93 | 0.50 | 3 |
| Other | (100) Phthalimide | 85-41-6 | SVOC | 7.1 ± 2.5 | 1.2 ± 0.42 | 0.37 | 1 |
| Other | (101) 1-Octadecanol | 112-92-5 | SVOC | 6.3 ± 4.3 | 0.57 ± 0.39 | 0.31 | 2 |
| Other | (102) 1-Tridecene | 2437-56-1 | SVOC | 0.6 ± 0.6 | 0.08 ± 0.08 | 0.27 | 11 |
| Other | (103) 2-Tridecanone | 593-08-8 | SVOC | 2.0 ± 1.2 | 0.25 ± 0.15 | 0.20 | 5 |
| Other | (104) Acetone | 67-64-1 | I/VOC | 430 ± 190 | 180 ± 80 | 0.23 | 10 |
| Other | (105) 2-Cyclopenten-1-one | 930-30-3 | I/VOC | 71 ± 22 | 21 ± 7 | 0.32 | 1 |
| Other | (106) Methyl Ethyl Ketone | 78-93-3 | I/VOC | 1700 ± 700 | 580 ± 250 | 0.26 | 4 |
| PAH | (107) Naphthalene | 91-20-3 | I/VOC | 130 ± 130 | 25 ± 25 | 0.32 | 3 |
| PAH | (108) 1-Methylnaphthalene | 90-12-0 | I/VOC | 25 ± 22 | 4.3 ± 3.8 | 0.25 | 3 |
| PAH | (109) 2-Methylnaphthalene | 91-57-6 | I/VOC | 39 ± 44 | 6.8 ± 7.6 | 0.30 | 3 |
| PAH | (110) 2-Methoxynaphthalene | 93-04-9 | SVOC | 1.0 ± 0.9 | 0.16 ± 0.13 | 0.33 | 1 |
| PAH | (111) Acenaphthylene | 208-96-8 | SVOC | 0.7 ± 1.7 | 0.11 ± 0.27 | 0.61 | 3 |
| PAH | (112) Fluorene | 86-73-7 | SVOC | 1.6 ± 1.3 | 0.24 ± 0.18 | 0.21 | 8 |
| PAH | (113) Phenanthrene | 85-01-8 | SVOC | 2.4 ± 2.1 | 0.33 ± 0.29 | 0.17 | 1 |
| PAH | (114) Pyrene | 129-00-0 | SVOC | 0.7 ± 0.9 | 0.09 ± 0.11 | 0.32 | 10 |
| THM | (115) Chloroform | 67-66-3 | I/VOC | 1900 ± 700 | 380 ± 150 | 0.33 | 1 |
| THM | (116) Bromoform | 75-25-2 | I/VOC | 9.0 ± 2.9 | 0.87 ± 0.28 | 0.28 | 1 |
| THM | (117) Dibromochloromethane | 124-48-1 | I/VOC | 6.3 ± 7.5 | 0.74 ± 0.88 | 0.45 | 5 |
| DCB | (118) p-Dichlorobenzene | 106-46-7 | I/VOC | 38 ± 41 | 6.3 ± 6.8 | 0.27 | 5 |
| DCB | (119) o-Dichlorobenzene | 95-50-1 | I/VOC | 11 ± 3.0 | 1.9 ± 0.5 | 0.36 | 10 |
| Siloxanes | (120) D4 | 556-57-2 | I/VOC | 89 ± 64 | 7.4 ± 5.3 | 0.29 | 8 |
| Siloxanes | (121) D5 | 541-02-6 | I/VOC | 370 ± 550 | 24 ± 36 | 0.32 | 8 |
| Solvents | (122) Isophorone | 78-59-1 | I/VOC | 11 ± 35 | 2.0 ± 6.2 | 0.21 | 11 |
| Solvents | (123) PCBTF | 98-56-6 | I/VOC | 590 ± 1490 | 80 ± 200 | 0.68 | 7 |

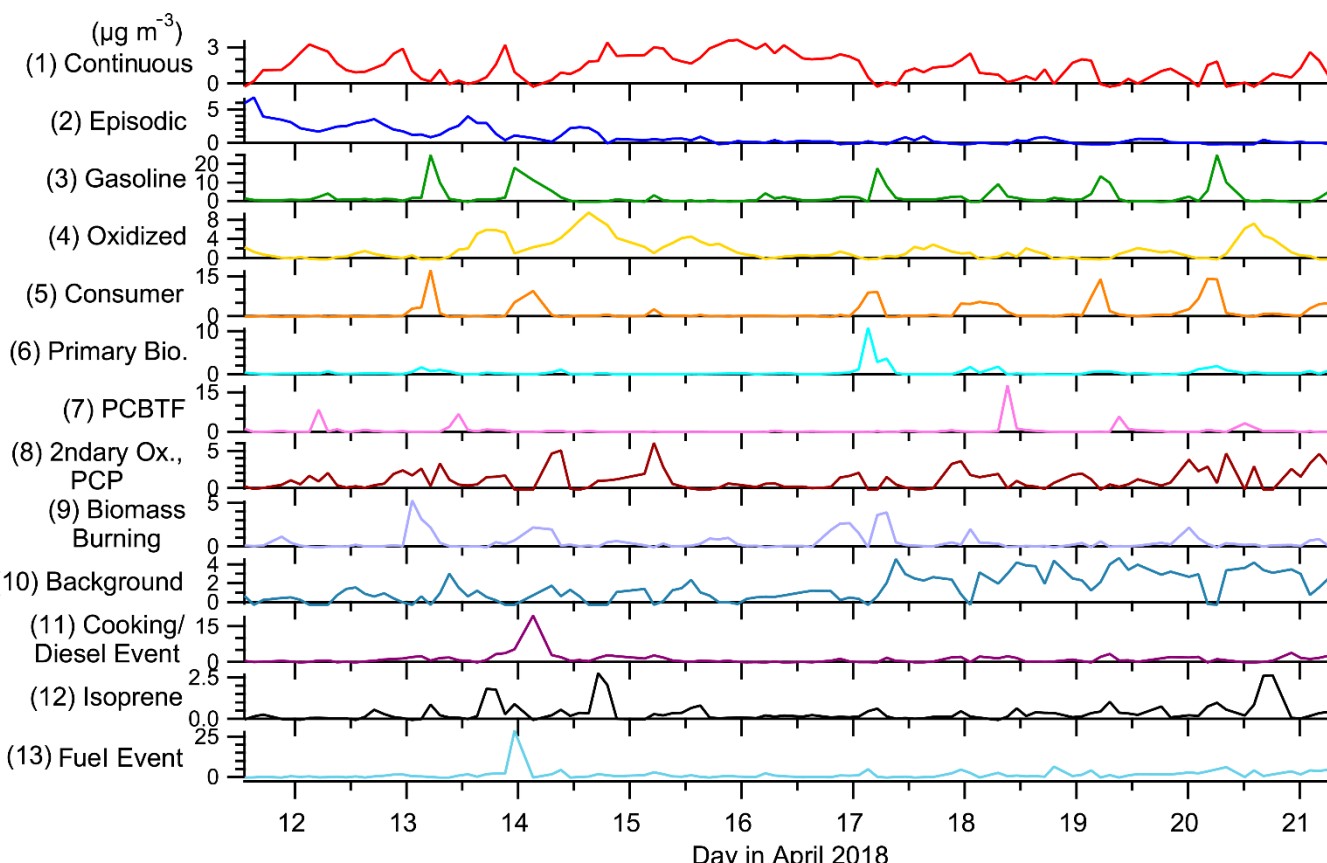


**Figure 1. 13 factor solution factor timelines. Bio. = Biogenic; PCBTF = Parachlorobenzotrifluoride; 2ndary Ox., PCP = Secondary Oxidation and Personal Care Products. Full factor names are in the main text (Sect. 3.1). Date labels are at midnight.**

The 13 factor solution has a $Q/Q_{exp}$ value of 0.79 which is close to 1, implying the data are neither overfitted nor underfitted (Ulbrich et al., 2009). Positive FPEAK rotations increase factor cross correlations and $Q/Q_{exp}$ and were therefore

not considered (Fig. S4 and Fig. S5). Negative FPEAK rotations slightly decrease factor cross correlations but increase $Q/Q_{exp}$. We chose to proceed with the unrotated solution because the improvement in factor cross correlations for negative FPEAK rotations was minor and because the unrotated solution minimizes $Q/Q_{exp}$.

Figure 1 shows the timelines for the 13 factor solution and Fig. 2 shows the corresponding factor profiles. The diurnal profiles of each of the factor timelines are plotted in Fig. 3, and relevant meteorological data and stacked factor

timelines are plotted in Fig. 4. Rose plots for periods of elevated concentration for each factor are presented in Fig. 5. The factors are presented in the order that they first appear in the PMF solution for increasing number of factor solutions, e.g., factor 9 in the 13 factor solution is approximately equivalent to the 9[th] factor in the 9 factor solution. The factors are (1) long-



lived and continuously emitted compounds, (2) episodic petrochemical source, (3) gasoline, (4) oxidized urban and temperature-driven emissions, (5) consumer products, (6) primary biogenic and diesel, (7) Parachlorobenzotrifluoride

(PCBTF), (8) secondary oxidation and persistent personal care product emissions, (9) biomass burning, (10) industrial and/or agricultural background and continuous combustion source, (11) early morning cooking/diesel event, (12) isoprene, and (13) possible jet fuel event.

Compounds measured on the I/VOC channel dominate by mass, representing 87 % of the total mass of the 123 included compounds. No factor had less than 50 % of its mass in VOCs and IVOCs (Fig. S6(a)), reflecting the fact that even

a small fractional contribution from a VOC or IVOC is significant compared to large fractional SVOC contributions due to the generally greater concentrations involved. SVOC mass was less evenly distributed between factors than I/VOC mass (Fig. S6(b)), controlled in part by stearic acid and bis(2-ethylhexyl) adipate (hereafter DEHA) contributions (Table 1).

### 3.2 Factor 1: Long-lived and continuously emitted compounds

Factor 1 is mainly long-lived, well-mixed species and species with continuous emissions. Chloroform and bromoform have

their highest mass fraction in this factor (33 % and 28 % respectively). Other well-represented compounds include 4-hydroxybenzoic acid (35 %), p-anisic acid (26 %), phthalimide (37 %), 2-cyclopenten-1-one (32 %), and 2-

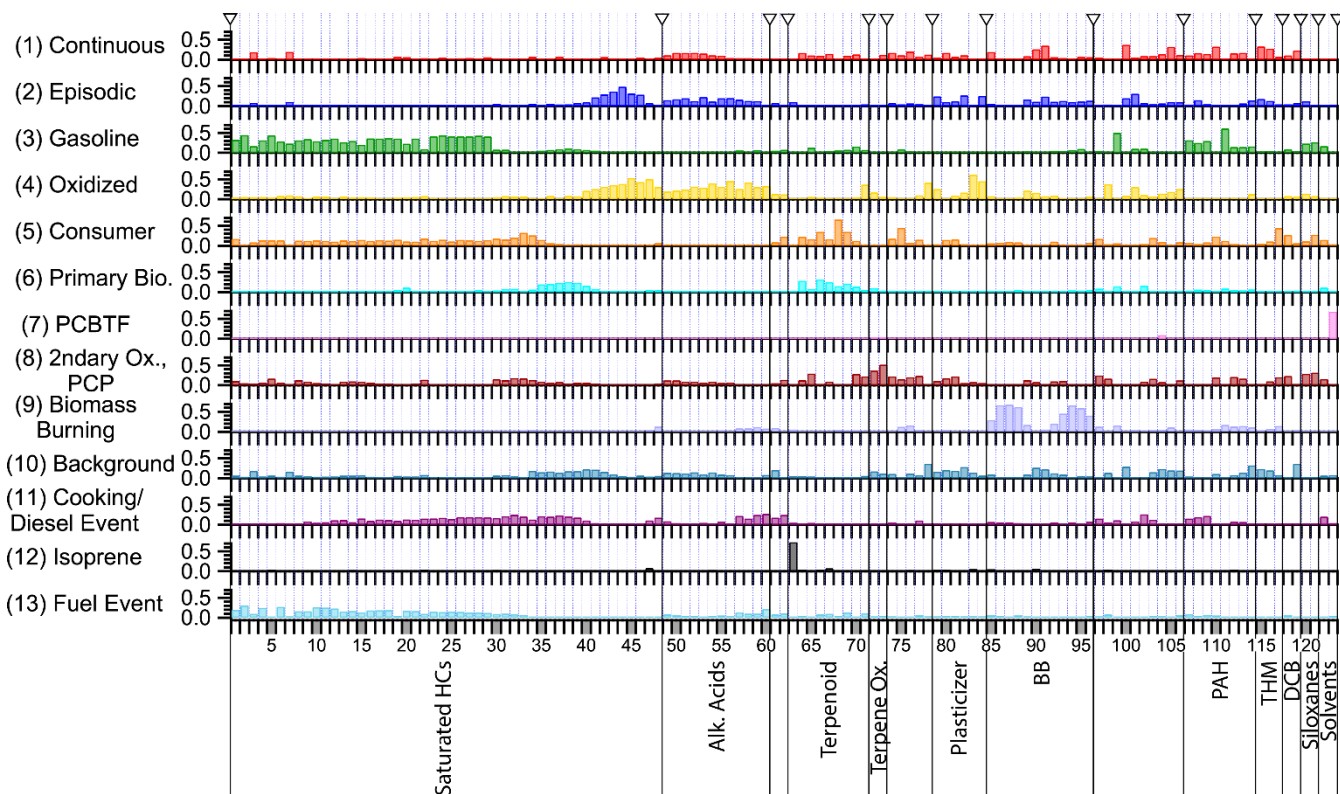

**Figure 2. 13 factor solution factor profiles. Compounds are ordered as in Table 1 using the indices assigned. Unlabeled groups are labeled "other" in Table 1.**





methoxynaphthalene (33 %). This factor is mildly elevated at night, but with high uncertainty, and compounds with a slightly opposite diurnal profile (e.g. 4-hydroxybenzoic acid, phthalimide) can also be found in this factor. When Factor 1 is elevated winds come from all directions, with a mild southwest preference (Fig. 5). This factor contains high fractional contributions from VOCs, IVOCs and SVOCs, representing a source profile that could not previously be obtained in a single instrument without the expanded volatility range afforded by cTAG.

Both chloroform and bromoform have atmospheric lifetimes of weeks or more (World Meteorological Organization et al., 2019), allowing them to be well-mixed in the troposphere and to therefore have stable concentrations unaffected by meteorological phenomena such as boundary layer height changes. Chloroform accounts for about 40 % of the total mass of Factor 1.

The other compounds present in this factor have the common characteristics of (1) being constantly present and (2) having a
weak to nonexistent diurnal variation, but their known sources are distinct and they are much more reactive than chloroform and bromoform. 4-hydroxybenzoic acid and p-anisic acid have been identified as tracers for burning of grasses (Simoneit, 2002) and have an estimated atmospheric lifetime with respect to OH ([OH] = 2 x $10^6$ molecules cm$^{-3}$ for this and every atmospheric lifetime estimate hereafter) of about a day (U.S. EPA, 2022). Phthalimide is a fungicide and insecticide degradation product produced in the processing of crops. It may also be formed from phthalic anhydride and primary amino
groups in the high temperature desorption process just before chromatographic separation, a possibility that we cannot rule out (Gao et al., 2019). Phthalimide has an estimated OH lifetime of a few hours (U.S. EPA, 2022). Atmospheric studies of 2-cyclopenten-1-one, which is naturally occurring and biologically significant (PubChem, 2022), and of

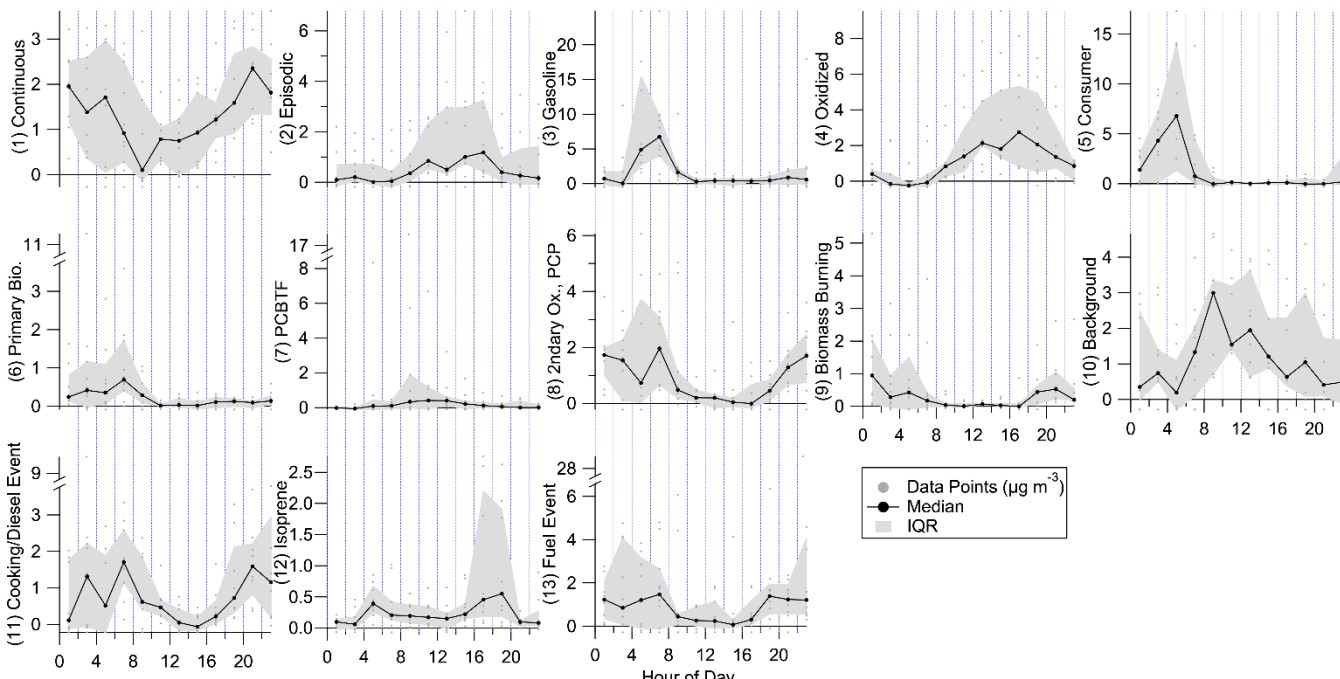

Figure 3. Diurnal profiles of the factor timelines in the 13 factor solution. IQR = Interquartile range.



2-methoxynaphthalene, which has industrial sources and uses (European Chemicals Agency, 2022), are lacking. The latter has an estimated lifetime with respect to reaction with OH of a few hours (U.S. EPA, 2022).

These relatively reactive compounds must have constant emission sources to produce timelines with so little variability. Because Factor 1 is associated with winds from all directions, the sources would need to be either hyper local or ubiquitous in the surrounding area. Agricultural activity may be able to account for 4-hydroxybenzoic acid, p-anisic acid,

phthalimide, and 2-cyclopenten-1-one.

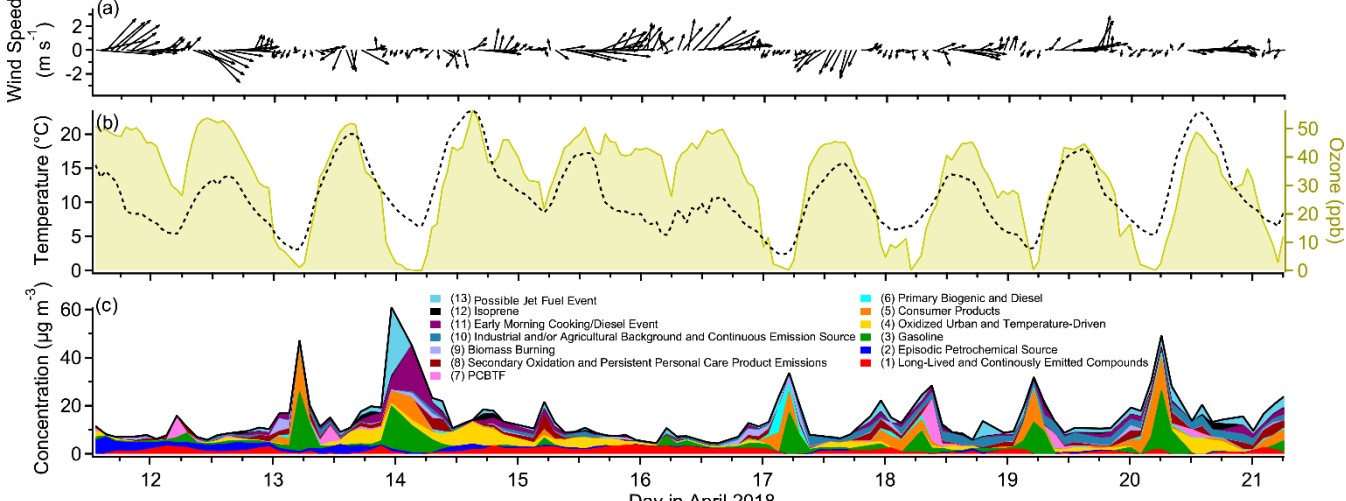

**Figure 4. (a) Time series of wind speed and direction, (b) time series of temperature (left axis) and ozone (right axis) measured at the Bay Area Air Quality Management District monitoring site, and (c) all factor timelines stacked. Date labels are at midnight.**

### 3.3 Factor 2: Episodic petrochemical source

Factor 2 is a transient source, elevated during the first few days of the measurement period and nearly nonexistent after that. It contains minor contributions from $C_{20}$–$C_{25}$ $n$-alkanes (mass fraction 20–50 %), 1-octadecanol (31 %) and some semivolatile phthalates (~25 % each). Winds tend to originate from the west when this factor is high (Fig. 5).

The heavy $n$-alkanes are split between Factors 2 and 4 (oxidized urban and temperature-driven emissions), with slightly different distributions; the Factor 2 $n$-alkanes skew slightly lighter, while Factor 4 includes contributions from $C_{26}$

and $C_{27}$ alkanes. $C_{20}$–$C_{27}$ $n$-alkanes are associated with petroleum-based products (Simoneit, 1999). (Biogenic sources of heavy alkanes exhibit a preference for alkanes with an odd number of carbon atoms over those with an even number (Simoneit, 1989), which is not observed in this dataset, confirming the fossil fuel origin of the $n$-alkanes.) Two possible sources in this study are motor oil (Caravaggio et al., 2007; Mao et al., 2009; Isaacman et al., 2012) and asphalt (Rogge et al., 1997; Khare et al., 2020), both of which exhibit temperature-dependent evaporation of alkanes in this carbon number

range. Since the alkane distribution is different between the two factors, it is reasonable to assume the elevated levels of some of these alkanes at the beginning of the campaign represent a different source than the regular afternoon maxima characteristic of Factor 4. The center of the $n$-alkane mass distribution in motor oil is typically at $C_{25}$ or greater, though



Isaacman et al. (2012) observed a peak at $C_{24}$ for motor oil aerosol particles due to preferential evaporation of lighter alkanes at low temperatures. Maximum fresh asphalt alkane emissions occur at $C_{14}$, though after 2 to 3 days emission is approximately constant across $C_{14} - C_{32}$ alkanes (Khare et al., 2020). Since the alkane distribution of asphalt emissions skews lighter, one possibility is that the elevated concentration of Factor 2 in the first few days is in part due to fresh paving nearby. However, the composition of motor oil is highly variable (Mao et al., 2009); the sources of alkanes in Factors 2 and 4 could be two different motor oils, for example, or could represent fresh vs older asphalt.

The semivolatile phthalates in this factor, dibutyl phthalate and benzyl butyl phthalate, are plasticizers used as coatings in manufactured goods and in building materials (Zota et al., 2014). Their emission exhibits temperature dependence (Fujii et al., 2003). In this study, their concentrations are moderately correlated with temperature ($r = 0.59$ for dibutyl phthalate and $r = 0.42$ for benzyl butyl phthalate), unlike the less volatile plasticizers bis(2-ethylhexyl) adipate and bis(2-ethylhexyl) phthalate, which correlate very strongly with temperature and mostly show up in Factor 4, and the more volatile phthalates associated with personal care products (see discussion for Factor 5). Again we may speculate a unique and temporary source for these two compounds at the beginning of this field campaign, plausibly related to construction activity that could also be responsible for *n*-alkane emissions from fresh asphalt.

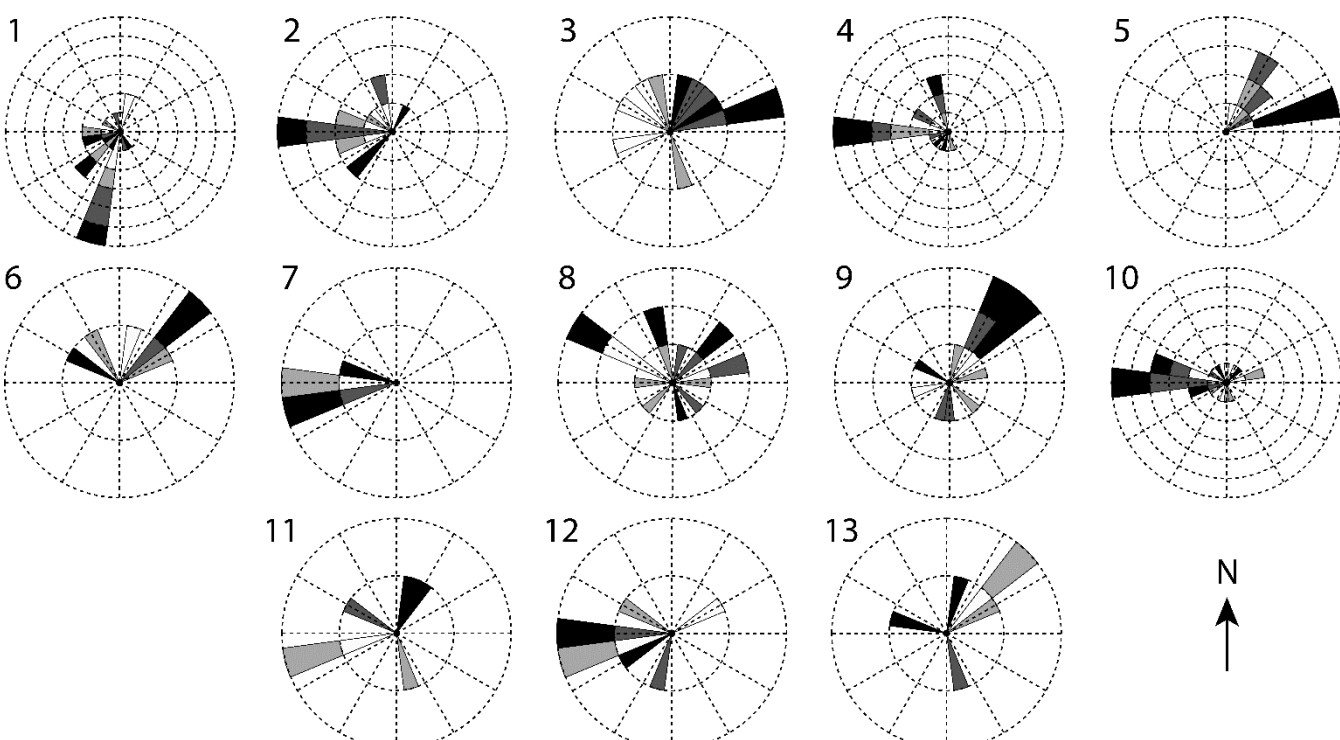

**Figure 5. Rose plots, showing the correlation of emissions and wind direction, of each factor using only data points where the concentration was elevated, defined as > 1 standard deviation above the mean factor concentration. Frequency of observations are represented by the length of each wedge, where each ring corresponds to one observation. Shading corresponds to quartiles of factor concentration (darker = greater concentration).**





1-octadecanol has vegetation/microbial (Nolte et al., 2002; Simoneit, 2006), biomass burning (Nolte et al., 2001), and anthropogenic (as detergent; Mudge et al., 2012) origins. 1-octadecanol does not appear in Factor 9 (biomass burning), though two short elevated concentration periods on the evening of 16 April and the early morning of 17 April coincide with 345 one of the biomass burning episodes. Given the non-persistent nature of Factor 2 and 1-octadecanol, an intermittent anthropogenic detergent source seems more likely than a biogenic source, which would likely emit continuously or at least regularly throughout the sampling period.

### 3.4 Factor 3: Gasoline

Factor 3 is designated as primary gasoline exhaust. Compounds with the highest mass fraction in this factor include 350 linear, branched and aromatic hydrocarbons with 6 to 10 carbon atoms and polycyclic aromatic hydrocarbons (PAHs), especially naphthalene and methylnaphthalenes. These are the main organic components of gasoline exhaust (Schauer et al., 2002b; Gentner et al., 2013). Palmitoleic acid also has a high mass fraction contribution, and octamethylcyclotetrasiloxane (D4) and decamethylcyclopentasiloxane (D5) are present. In periods of high concentration for this factor, wind speeds were low (about 1 m s$^{-1}$ or lower) and wind directions were variable, with a slight preference for northeastern origin (Fig. 5).

This factor exhibits a strong diurnal pattern with sharp peaks in concentration in the early morning (between 5 and 7AM local standard time) and near zero loading at other times of day. This factor strongly correlates with NO$_x$ (Pearson's r = 0.88) and anti-correlates with ozone (r = -0.66). With over 80 % of Livermore workers commuting by private vehicle or carpool (United States Census Data, 2022), the early morning elevated concentration is likely commute-related. Our hypothesis for the exhibited diurnal profile is as follows. First, emissions of both hydrocarbons and NO$_x$ from automobile 360 tailpipes are low overnight but increase in the early morning hours as morning commutes begin with cold engine starts. Drozd et al. (2016) found that emissions from cold starts dominate non-methane organic gas emissions from light duty vehicles, with over 45 km of driving required to match the initial cold start emissions for over 97 % of light duty vehicles in use in the United States today. The morning peaks are correlated with, and enhanced by, cooler temperatures and generally slower wind speeds (Fig. 4). The shallow planetary boundary layer overnight and absence of sunlight allow NO$_x$ and 365 hydrocarbons to build up, raising measured concentrations. As the day progresses, the boundary layer rises and wind speeds increase, diluting species near ground level, and photochemistry begins, reacting away hydrocarbons and NO$_x$ and producing ozone. Late afternoon and evening return commutes have greatly reduced emissions because vehicle catalytic converters are already hot by the time drivers return to the predominantly residential area surrounding the sampling site. Higher ambient temperatures in the afternoon also contribute to reduced cold start emissions and greater wind speeds and more rapid 370 oxidative loss prevent what emissions are still produced from building up. In the late evening and night, emissions remain low and thus concentrations also remain low despite the low boundary layer and lack of photochemistry.

Factors 5 and 6 and many marker compounds share this general pattern of elevated concentrations exclusively in the early morning hours. They are governed by the same large-scale atmospheric processes. Thus while differences exist which allow them to be separated by the PMF model, which will be discussed in the descriptions for those factors, their overarching


similarity means the sources are not separated perfectly. For example, palmitoleic acid is primarily a tracer of cooking (Rogge et al., 1996; Robinson et al., 2006), yet has its highest mass fraction in this factor, while other cooking tracers appear in other factors. Together with the overall similarity in variability between gasoline and cooking markers, palmitoleic acid's low and noisy signal may have caused it to be placed in this factor primarily by the model, even though it does not originate from the same pollution source.

380       D4 and D5 siloxanes have substantial fractions of their mass in this factor (23 % and 27 % respectively). D5 has been established as a tracer compound for personal care product emissions (Horii and Kannan, 2008; Wang et al., 2009; Coggon et al., 2018). It is also prominent (mass fraction of 28 %) in Factors 5 and 8, which both contain fragrance compounds commonly found in consumer products, including personal care products. The fact that D5 is present in Factors 3, 5 and 8 while the fragrance compounds are only present in Factors 5 and 8 could be reflective of different emission rates

from those products. D5 is found in the greatest concentrations in antiperspirants (Wang et al., 2009); once applied, while emissions are highest immediately after application, it evaporates over the course of several hours (Montemayor et al., 2013) and has even been directly measured in an engineering classroom in the afternoon (Tang et al., 2015) and outside an automobile in cabin fan exhaust from human occupants who had applied D5-containing personal care products earlier that day (Coggon et al., 2018). Evaporation of monoterpenes and monoterpenoids, which comprise the vast majority of the mass

of fragrance compounds measured in this study, is likely to happen much faster due to their greater volatility (Hazardous Substances Data Bank, 2022). A study of evaporation of 300 μL of essential oils indoors found that most VOC mass, including most monoterpene mass, was emitted during the first 30 min (Su et al., 2007).

      D4 is mainly indicative of adhesive and pesticide use (Gkatzelis et al., 2021) but is also found in consumer products (Horii and Kannan, 2008; Wang et al., 2009), though it is not present in Factor 5, the consumer product factor. If the D4

measured in Livermore is from consumer products, it is not clear why the peak concentration coincides with the gasoline markers, later in the day than the peak of the other consumer product compounds.

      It is likely that some mass from the diesel emission source is in this factor. Black carbon, which originates almost entirely from diesel emissions (Gentner et al., 2013), correlates best with this factor (r = 0.84), suggesting some of the variability in this factor is driven by diesel exhaust emissions of compounds found in both gasoline and diesel exhaust.

**3.5 Factor 4: Oxidized urban and temperature-driven emissions**

      Factor 4 represents aged urban emissions and temperature-driven emissions. This factor includes heavy *n*-alkanes (C$_{20}$–C$_{27}$) as well as the *n*-alkanoic acid series (C$_8$–C$_{18}$), with approximately 30–50 % of the mass of each compound in these two classes included in this factor. Several other compounds are also present, including DEHA (62 %), bis(2-ethylhexyl) phthalate (hereafter DEHP; 46 %), phthalic anhydride (42 %), 4-nitrophenol (38 %), aromadendrene (37 %) and azelaic acid

(33 %). This factor contains the largest fraction of semivolatile mass of any of the factors (Fig. S6(a)), largely due to the presence of DEHA and the fatty acid series. Factor 4 correlates strongly with temperature (Pearson's r = 0.82), peaking in the mid to late afternoon. Winds are moderately strong and from the west when this factor is elevated (Fig. 5). The smooth



variation in concentration of Factor 4 suggests a regional source rather than a local one, which would likely display greater inter-hourly variability in concentration and greater sensitivity to small changes in wind direction.

DEHA and DEHP are both semivolatile plasticizers found in polyvinyl chloride, commonly found in building materials (Liu and Little, 2012; Shi et al., 2018; Bui et al., 2016). Like Factor 4, these two compounds correlate strongly with temperature (r = 0.83 for DEHA and r = 0.73 for DEHP), consistent with temperature-dependent emission which has been observed in controlled studies (Clausen et al., 2012; Fujii et al., 2003; Liang and Xu, 2014). The heavy alkanes are discussed in detail in the Factor 2 (episodic petrochemical source) section. They are known to originate from evaporative, temperature-

dependent sources such as motor oil and asphalt, consistent with the temporal profile of Factor 4.

The remaining compounds represented in this factor are not likely to originate from evaporative sources, but given the consistent winds from the west could represent oxidized emissions transported from the east and south Bay Area. Phthalic anhydride is a secondary product of the photooxidation of naphthalene and phthalic acid has been found in secondary organic aerosol formed from naphthalene photooxidation (Chan et al., 2009; Kleindienst et al., 2012; Wang et al.,

2007). During the study period, conditions for optimal photooxidation (i.e., high solar flux) coincided with periods of high temperature, causing these distinct source categories to appear in the same PMF factor.

In urban and suburban areas, the n-alkanoic acid homologous series is most often ascribed to cooking emissions (Schauer et al., 2002a; Robinson et al., 2006; Allan et al., 2010; Mohr et al., 2012; Yao et al., 2021), but biomass burning, motor vehicle exhaust and road dust can all contribute (Schauer et al., 1996; Rogge et al., 1996). The fatty acids also

originate from terrestrial microbial activity (Simoneit and Mazurek, 1982) and marine phytoplankton (Kawamura et al., 2003), sources that tend to dominate in remote areas (Kawamura et al., 2003; Cahill et al., 2006; Fu et al., 2014; Boreddy et al., 2018). Alkanoic acids from cooking exhibit a distinct diurnal profile, with elevated concentrations around dinner time and occasionally another similar peak around lunchtime (Allan et al., 2010; Mohr et al., 2012; Dall'Osto et al., 2015; Yao et al., 2021). In this study, palmitic ($C_{16}$) and stearic ($C_{18}$) acids, which are emitted from meat cooking (Rogge et al., 1991), do

show occasional evening elevated concentrations (not captured in the Factor 4 profile), but the other alkanoic acids do not. The defining feature that likely caused the fatty acids to be grouped in this factor is the period of sustained elevated concentrations between the evenings of 13 April and 15 April, with almost no diurnal sensitivity (Fig. 6). This temporal profile is inconsistent with local emissions from the sources mentioned above, but an elevated regional background transported from urban areas to the west could explain the variability. Azelaic acid, the $C_9$ dicarboxylic acid, has similar

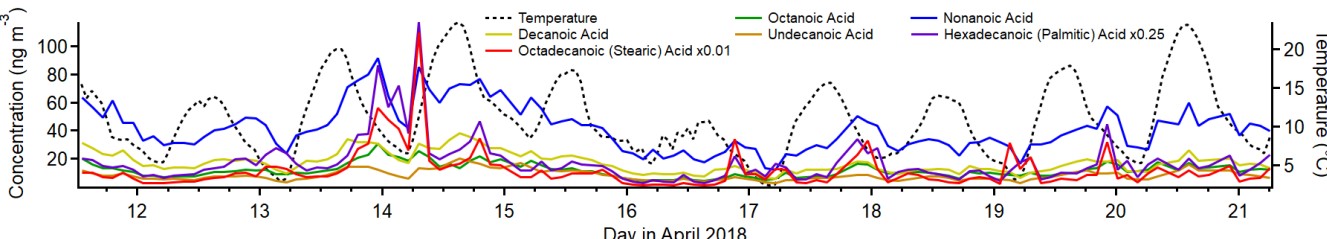


**Figure 6. Timelines of select alkanoic acids (left axis) and temperature (right axis) for the study period.**



sources to the n-alkanoic acids (Kawamura and Bikkina, 2016) and a similar temporal profile to palmitic and stearic acids, including brief evening spikes in concentration likely from cooking and the two-day period of elevated concentration.

440       The timeline of the sesquiterpene aromadendrene exhibits the same period of sustained elevated concentration as the alkanoic acids and azelaic acid. Sesquiterpene emissions from plants are temperature dependent (Duhl et al., 2008; Bouvier-Brown et al., 2009). Emission outpaces loss on some days despite aromadendrene's very high reactivity with ozone (Pollmann et al., 2005) and the hydroxyl radical (Ng et al., 2007).

      4-nitrophenol is strongly represented in this factor (38 % mass fraction), while its isomer 2-nitrophenol does not contribute at all. Both nitrophenols are emitted during motor vehicle combustion (Nojima et al., 1983; Tremp et al., 1993)
and in some industrial manufacturing (Harrison et al., 2005b). Biomass burning also leads to production of 4-nitrophenol and possibly 2-nitrophenol (see Factor 9 discussion). Secondary formation of nitrophenols occurs from photooxidation of aromatic hydrocarbons such as benzene, toluene, phenol and cresol (Harrison et al., 2005b). Primary emissions from vehicles and secondary formation are thought to be the most important sources of nitrophenols in polluted urban environments (Inomata et al., 2013; Lu et al., 2019).

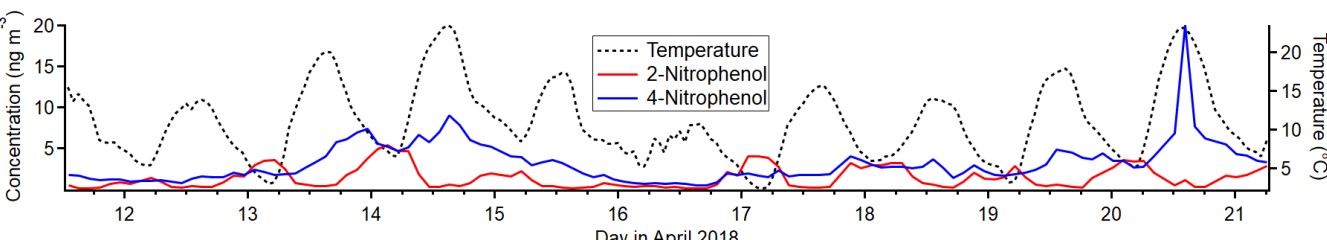


**Figure 7. Timelines of nitrophenol isomers (left axis) and temperature (right axis) for the study period.**

      4-nitrophenol is observed in greater concentrations than 2-nitrophenol in this study both on average and at each data point except for about 10 nighttime samples (Fig. 7), even in the gas phase. This is in contrast to other studies (Lüttke et al., 1999; Cecinato et al., 2005), though most modern studies did not distinguish between the two isomers (Yuan et al., 2016;
Cheng et al., 2021; Wang et al., 2020; Salvador et al., 2021) or did not quantify 2-nitrophenol (Kitanovski et al., 2012; Wang et al., 2017, 2018, 2019b). Nitrophenols from vehicle exhaust show a mild preference for the ortho isomer (Nojima et al., 1983; Tremp et al., 1993), and gas-phase secondary formation favors 2-nitrophenol as well (Harrison et al., 2005a). Liquid-phase secondary formation is likely to be an important production mechanism for nitrophenols, but the ratio of isomer production for this process is unknown (Harrison et al., 2005a). Observations of particle-phase nitrophenol show a strong
preference for the para isomer (Lüttke et al., 1999; Harrison et al., 2005b), likely due to its much lower vapor pressure and much higher Henry's law constant (Sander, 2015; U.S. EPA, 2022), but 4-nitrophenol is observed predominantly in the gas phase in this study, consistent with other studies (Yuan et al., 2016; Cheng et al., 2021). There are several loss processes of gas-phase nitrophenol, with photolysis thought to be the most important (Vione et al., 2009; Yuan et al., 2016), though comparison of gas-phase photolysis between 2NP and 4NP is not currently feasible (Chen et al., 2011; Sangwan and Zhu,
465  2018).





Our data suggest 2-nitrophenol is emitted and concentrated in the nighttime hours and lost in the daytime, while 4-nitrophenol is either not lost during the daytime or has an additional daytime source that 2-nitrophenol lacks, such as regional transport from the Bay Area to the west. The state of the existing literature leaves both possibilities open, and without more information it is not possible to determine what is causing the difference in profiles between these two isomers.

**3.6 Factor 5: Consumer products**

Factor 5 is designated as consumer product emissions. It consists of limonene and other biogenic compounds, D5 siloxane, brominated and chlorinated hydrocarbons and small contributions from volatile diesel markers. Of the biogenic compounds present, monoterpenes, methyl salicylate and α-isomethyl ionone are well represented while isoprene, the monoterpenoids camphor and eucalyptol, the sesquiterpene aromadendrene, and α-pinene oxidation products pinic and
pinonic acid are not. When the concentration of this factor is high, winds come exclusively from the northeast, though wind speeds are low (< 1 m s$^{-1}$) (Fig. 5). The diurnal profile of this factor qualitatively resembles that of Factor 3 (gasoline), with strongly elevated concentrations in the early morning hours exclusively. Peak consumer products emissions occur slightly earlier than gasoline emissions on average, from about 04:00 to 06:00 LT (Fig. 3).

As a monoterpene, limonene is often considered to originate from biogenic sources, but limonene is also used
ubiquitously in personal care products and cleaning products as a fragrance (Logue et al., 2011; Steinemann et al., 2011). Coggon et al. (2021) showed that fragranced products are the dominant source of limonene in some urban areas. While limonene is the monoterpene typically found in the highest concentrations in fragranced consumer products, the other monoterpenes included in this PMF analysis (α-pinene, β-pinene, camphene and 3-carene) are also common components of consumer products (Steinemann et al., 2011; Steinemann, 2015). The early morning spikes in concentration just before
gasoline markers become elevated is consistent with consumer product use by individuals preparing for their day before starting their cars to drive to work. In contrast, studies reporting monoterpene concentrations in rural or remote locations, where biogenic emissions likely dominate, show sustained elevated concentrations throughout the night (Bouvier-Brown et al., 2009), not just early morning.

D5 is also prominent in this factor and, as stated in the discussion of Factor 3, is a tracer for personal care product
emissions. Methyl salicylate, as the primary component of wintergreen oil, is also used as a fragrance compound (Lapczynski et al., 2007). α-isomethyl ionone is naturally found in Brewer's yeast and emitted during fermentation (Loscos et al., 2007), but is also commonly produced synthetically and found in cosmetics and personal care products (del Nogal Sánchez et al., 2010). There are three breweries about 1.1 km to the southeast of the sampling site and one about 3.5 km to the northeast; given their distance, the low northeastern winds and the presence of other personal care product fragrance
compounds in this factor it is most likely α-isomethyl ionone measured at the sampling site also originates from personal care products. Methyl salicylate and α-isomethyl ionone are low-volatility IVOCs, measured on the SVOC channel of cTAG. Their presence in this factor contributes to a more comprehensive source profile for consumer product emissions than could be obtained with a VOC-only measurement focus.



*p*-Dichlorobenzene is an industrial chemical used as a deodorant and insect repellent, especially in mothballs
(ATSDR, 2011). It is widely available as a consumer product and is typically placed in enclosed spaces with clothes
vulnerable to moth damage (Chin et al., 2013). Its inclusion in this factor and Factor 8 could be explained by morning
occupant activity that disturbs and ventilates spaces where p-dichlorobenzene is placed, temporarily increasing their
emission to the rest of the indoor environment and, in turn, outdoors. It is also possible that general household activity
correlates with VOC exchange from indoor residences to outdoors, independent of whether occupants are interacting with *p*-
dichlorobenzene-containing substances. Further highly time resolved studies are needed to assess time-of-day exposure and
transport to the outdoors.

Dibromochloromethane is a disinfection byproduct found in chlorinated tap water (Krasner et al., 1989). It is also
produced from marine macroalgae (Manley et al., 1992; Sturges et al., 1992; Carpenter and Liss, 2000), the more important
source globally (World Meteorological Organization et al., 2019), and is well-mixed in the troposphere, with a lifetime of 70
days (World Meteorological Organization et al., 2019). One likely source of this compound above the regional background is
the outdoor swimming pool 10 m to the northeast, which was closed to the public for the season but was nonetheless kept
filled. Trihalomethane emissions from commercial pools have been extensively documented (Fantuzzi et al., 2001; Zwiener
et al., 2007; Richardson et al., 2010; Righi et al., 2014; Westerlund et al., 2019). While pool emissions may be responsible
for elevated concentrations throughout the nighttime when dilution is low, the spikes in concentration in the early morning
hours are more likely related to human activity. Residential showering has been shown to increase the concentration of
dibromochloromethane in the bathroom air by 10 µg m$^{-3}$ or more, depending on its initial concentration in the tap water
(Kerger et al., 2000). Once ventilated outside, it is plausible that with enough temporally coincident showers in nearby
residences the concentration measured at the sampling site would increase by the observed 5–10 ng m$^{-3}$ in the early morning
hours.

The C$_{10}$–C$_{16}$ *n*-alkanes contribute between 15 and 32 % of their mass to this factor. These compounds come chiefly
from gasoline and diesel exhaust or fuel evaporation (Schauer et al., 1999b, 2002b; Gentner et al., 2013; Drozd et al., 2021),
but use of petroleum distillates in some consumer products is another contributor (McDonald et al., 2018). A consumer
product source would be the most consistent with the temporal variability and composition of this factor.

### 3.7 Factor 6: Primary biogenic and diesel

Factor 6 is designated primary biogenic and diesel. Three out of the five monoterpenes measured are shared
between this factor and the consumer products factor (camphene, mass fraction 29 %, α-pinene, 32 %, and β-pinene, 25 %).
The only other major constituents of Factor 6 are a narrow range of semivolatile *n*-alkanes (C$_{16}$–C$_{19}$) and pristane and
phytane, contributing about 20 to 30 % of their mass. Like Factors 3 (gasoline) and 5 (consumer products), Factor 6 is
elevated exclusively in the early morning hours. However unlike those two factors, the majority of the signal from Factor 6
is confined to one event in the early morning hours of 17 April. Winds were calm (<1 m s$^{-1}$) and from the northeast during
this event (Fig. 5).




While most of the measured monoterpene mass is likely due to consumer product emissions (see Factor 5 discussion), the personal care product tracer D5 contributes no mass to Factor 6. The peak of the early morning event on 17 April is at 3AM, two hours earlier than the typical diurnal maximum for Factor 5. Therefore this factor is more likely to represent concentrated local biogenic emissions.

The $C_{10}$–$C_{19}$ alkanes are mainly split between this factor, Factor 5 and Factor 11 (early morning cooking/diesel event). While Factor 11 contains contributions from $C_{10}$–$C_{19}$ alkanes as well as pristane and phytane, which are expected from diesel exhaust or evaporative emissions (Schauer et al., 1999b; Gentner et al., 2013), Factor 5 only contains contributions from the lighter alkanes in this range ($C_{10}$–$C_{16}$) and Factor 6 from the heavier ones ($C_{16}$–$C_{19}$, pristane, phytane). The event on 17 April that defines Factor 6 is relatively enriched in the heavier alkanes within this carbon number range, leading to splitting between factors. The Factor 5 lighter alkane group could come from consumer product use as mentioned in the Factor 5 discussion.

### 3.8 Factor 7: Parachlorobenzotrifluoride (PCBTF)

Factor 7 consists nearly exclusively of *p*-chlorobenzotrifluoride (PCBTF), with 68 % of that compound's mass present in this factor. No other compound has more than 10 % of its mass in this factor. The timeline is extremely episodic, with brief (single data point, or < 3.5-hour duration) concentration spikes on some days around noon. Winds are exclusively from the west when this factor's concentration is high (Fig. 5).

PCBTF is a volatile chemical product used exclusively in solvent-based coatings (Stockwell et al., 2021; Gkatzelis et al., 2021). Coatings are defined in emission inventories as paints, varnishes, primers, stains, sealers, lacquers, and any solvents associated with coatings (Stockwell et al., 2021). They tend to be associated with construction projects (McDonald et al., 2018) and Gkatzelis et al. (2021) found that concentrations of PCBTF specifically correlated poorly with population density, which is consistent with industrial rather than individual consumer use.

The specific source of PCBTF in this study is unknown. Though the direction of the source is well-defined, winds originated from the west for the entire morning and afternoon on the days when PCBTF was detected, not just during periods of elevated concentration of PCBTF (Fig. 4), suggesting the emissions themselves are intermittent rather than being continuous but only sampled intermittently. The lifetime of PCBTF in the atmosphere with respect to reaction with OH is over 20 days (Gkatzelis et al., 2021), so reactive loss is not expected to affect the measured concentration.

### 3.9 Factor 8: Secondary oxidation and persistent personal care product emissions

Factor 8 is designated as secondary oxidation products and semivolatile personal care product emissions. This factor contains moderate (≈25 % mass fraction or greater) contributions from 11 different compounds, with 1st generation α- and β-pinene oxidation products pinonic acid and pinic acid contributing the highest mass fractions (52 % and 36 % respectively). The factor concentration is elevated throughout the nighttime hours when ozone is present and lower at night



when ozone is low as well as midday. Winds come from all directions when this factor's concentration is high, with a slight preference for the northwest and northeast (Fig. 5), but wind speeds are low (< 1.8 m s$^{-1}$ for all but one data point).

565   Pinic and pinonic acids match the variability of Factor 8 the most closely. Both compounds are both secondary oxidation products and precursors of further oxidized species, with gas phase lifetimes of only a few hours under typical conditions but with low enough volatility to partition into the aerosol phase, which extends and adds uncertainty to their atmospheric lifetimes (Donahue et al., 2012). In general, concentrations are highest at night and mid-morning, and low in the middle to late afternoon. In the afternoon, even though ozone and α- and β-pinene are present, the concentrations of pinic

570 and pinonic acid dip, likely due to the combined effects of continued oxidation and physical dilution. At night, one of two scenarios occurs. When ozone remains high throughout the night (12, 15 and 16 April in this data set), these two acids build up under the shallow boundary layer. When ozone is low at night or early morning, local minima in pinic and pinonic acid concentrations are observed, leading to a bimodal diurnal trend. This trend is noisier and less obvious (though still present) in the Factor 8 timeline due to the contributions of other compounds not driven by the same chemistry.

575   In addition to pinic and pinonic acids, several compounds used in personal care products are either split between this factor and Factor 5 (consumer products) or are predominantly present in this factor. D5 siloxane's mass is approximately equally split between Factors 3 (gasoline), 5 and 8. Other compounds present in this factor that are used in personal care products include camphor (Drugsite Trust, 2021), eucalyptol (Medcraft and Schnell, 2016), benzophenone (Anderson and Castle, 2003; Downs et al., 2021; U.S. EPA, 2022), dimethyl and diethyl phthalates (Zota et al., 2014) and methyl salicylate

580 (Lapczynski et al., 2007), with the latter having approximately equal mass in Factors 5 and 8. *p*-Dichlorobenzene and dibromochloromethane, which are not from personal care products but are linked to morning indoor residential activity, also share mass between Factors 5 and 8.

   When comparing compounds predominantly present in one factor or the other, more volatile, less oxygenated compounds are represented in Factor 5, while oxygen-containing IVOCs and larger, semivolatile compounds, many of them

585 oxygen-containing, are found in Factor 8. Factor 8 species tend to have longer atmospheric lifetimes as well, with camphor (Reissell et al., 2001), eucalyptol (Corchnoy and Atkinson, 1990), methyl salicylate (Ren et al., 2020), benzophenone (U.S. EPA, 2022), dimethyl phthalate (Han et al., 2014), p-dichlorobenzene (Atkinson and Arey, 1993), dibromochloromethane (World Meteorological Organization et al., 2019) and D5 siloxane (Navea et al., 2011) all having lifetimes of over 1 day when exposed to typical hydroxyl radical concentrations ([OH] = 2 * 10$^6$ molecules cm$^{-3}$), and likely even longer lifetimes in

590 practice when the gas-particle partitioning of the semivolatile organics is taken into account (Kroll and Seinfeld, 2008; Cousins and Mackay, 2001). In contrast, the compounds in Factor 5 associated with personal care product emissions that are not also present in this factor (the monoterpenes) have atmospheric lifetimes of a few hours at most (Atkinson and Arey, 2003). This is consistent with the diurnal profiles of the two factors: long-lived personal care product compounds, while they may be emitted in the early morning, are not fully reacted away during the day and get reconcentrated under the boundary

595 layer in the evening, leading to elevated concentrations throughout the night. Short-lived compounds are only observed during and shortly after their morning emission.



D4 siloxane has a somewhat greater mass fraction in Factor 8 (29 %) than in Factor 3 (23 %). Like the compounds split between factors 5 and 8, D4 has a long atmospheric lifetime (11 days; Navea et al., 2011), likely leading to elevated concentrations outside of typical emission times, unlike the gasoline markers that make up the bulk of the mass of Factor 3. 2-Nitrophenol contributes moderately to Factor 8 (mass fraction 24.8 %). Possible sources are discussed in the Factor 4 and Factor 9 descriptions.

### 3.10 Factor 9: Biomass burning

Factor 9 represents primary biomass burning emissions. The top mass fraction contributors, in order, are galactosan, levoglucosan, syringaldehyde, mannosan, syringic acid, vanillic acid, 4-nitrocatechol, and furfural, all tracers of biomass burning (Simoneit et al., 1999; Simoneit, 2002; Bertrand et al., 2018; Finewax et al., 2018). This factor is consistently elevated exclusively in the nighttime hours. Winds are light (< 1.5 m s$^{-1}$ except for one data point) and come from all directions, with a mild northeast preference, when Factor 9 is elevated (Fig. 5).

Wood burning in residences for heat or recreation is a significant source of pollution in the Bay Area, with 25 % of primary PM$_{2.5}$ emissions attributed to residential wood burning annually (Kniss et al., 2017). That fraction rises to 33 % or more between November and April (Bay Area Air Quality Management District, 2012; Bhattacharyya, 2022). With average evening temperatures (19:00 to 21:00 LT) of 10.4 °C during the sampling period, some residential wood burning activity is anticipated. While we expect wood burning to occur only early in the nighttime hours before residents go to sleep, the lower boundary layer throughout the night traps emissions, keeping concentrations elevated.

Some biomass burning markers included in this analysis did not show up in Factor 9. 4-hydroxybenzoic acid and p-anisic acid are tracers for burning of grasses (Simoneit, 2002), an unlikely fuel source for residential fireplaces. 4-nitrophenol has been observed in biomass burning plumes or otherwise attributed to biomass burning (Hoffmann et al., 2007; Mohr et al., 2013; Wang et al., 2017, 2020) and is likely formed secondarily within hours inside the biomass burning plume (Mason et al., 2001). 2-nitrophenol, to our knowledge, has either not been detected in biomass burning plumes (Hoffmann et al., 2007), or was simply not measured or not distinguished from the other isomer (Mohr et al., 2013; Wang et al., 2017; Wang et al., 2020). However, biomass burning is thought to be a minor source for 4-nitrophenol in urban areas (Harrison et al., 2005b; Li et al., 2016), where motor vehicle combustion is responsible for primary emissions of nitrophenols as well as precursors to their secondary formation (Tremp et al., 1993; Harrison et al., 2005b), which also plays a major role in their formation (Lüttke et al., 1997, 1999; Harrison et al., 2005b; Yuan et al., 2016; Cheng et al., 2021). Further discussion on the sources and timelines of the nitrophenol isomers can be found in the Factor 4 description.

### 3.11 Factor 10: Industrial and/or agricultural background and continuous combustion source

Factor 10 is elevated in the latter half of the campaign (17 April late morning onwards) with maximum concentrations in the late morning and early afternoon hours but weak diurnal variability. Its main constituents include *o*-dichlorobenzene (mass fraction 36 %), phthalic anhydride (36 %), the PAH pyrene (32 %), high volatility phthalates (15–30





%) and grass burning tracers 4-hydroxybenzoic acid (22 %) and p-anisic acid (26 %), with a small contribution from $C_{15}$–$C_{21}$
alkanes, pristane and phytane (14–22 %). Winds come from all directions when this factor is elevated, but mostly from the
west (Fig. 5). This factor appears to be part of the regional background.

*o*-Dichlorobenzene is used in the production of herbicides and dyes and as a solvent (Meek et al., 1994; ATSDR,
2006) and lasts for weeks in the atmosphere (ATSDR, 2006). It contributes to Factor 1 (long-lived and continuously emitted
compounds) as well; as such, it is likely part of the regional background that could include industrial and/or agricultural
sources. It is not found in consumer products like its isomer *p*-dichlorobenzene.

PAHs would be expected to contribute to this combustion-related factor, because they are formed from incomplete
combustion (Ravindra et al., 2008). Unlike the other PAHs measured, which exhibit nighttime enhancement and are mostly
split between Factors 3, 8 and 9, pyrene shows only minor nighttime enhancement even though it has an atmospheric
lifetime with respect to reaction with OH of only a few hours (Atkinson and Arey, 1994), suggesting a continuous source. It
is unclear why pyrene's diurnal profile differs from the other PAHs.

Factor 10 and Factor 1 have several compounds in common, such as the grass burning tracers, trihalomethanes,
phthalimide, *o*-dichlorobenzene and to a lesser extent phthalates. They also share weak diurnal variability, though the
variability they do have is opposite. However, Factor 1 completely lacks any contribution from pyrene and the diesel
markers. It seems likely that both Factor 1 and Factor 10 represent some stable background level of long-lived or
continuously emitted compounds spanning the volatility range from VOCs to SVOCs, but that Factor 10 includes an
additional continuous combustion related source that was absent in the beginning of the campaign.

### 3.12 Factor 11: Early morning cooking/diesel event

Factor 11 consists nearly exclusively of a 14 April 03:00 spike in concentration of some cooking and diesel tracers
as well as a few other compounds. It includes 20–30 % of the mass of cooking tracers palmitic acid, stearic acid, azelaic acid
and nonanal and 10–25 % of the mass of the diesel tracers $C_{10}$–$C_{19}$ alkanes, pristane and phytane. 1-Tridecene and
isophorone have their highest mass fraction in this factor (27 % and 21 % respectively). Winds were calm ($\leq$ 1 m s$^{-1}$) and
from the northeast during this event (Fig. 5).

Palmitic acid, stearic acid and azelaic acid are cooking tracers as discussed in the Factor 4 description, though the
concentration variations captured in that factor are more likely to come from a difference source. Nonanal has biogenic
(Kirstine et al., 1998) and secondary (Atkinson, 2000; Fruekilde et al., 1998; Moise and Rudich, 2002) sources, but is also
emitted in significant quantities from cooking (Rogge et al., 1991; Schauer et al., 1999a, 2002a), the likely source for this
event. With northeast winds, one or more of the restaurants in the shopping center 100 m to the north could be the source of
the cooking and diesel activity.

Isophorone is a solvent for resins, polymers, wax, oil, pesticides, paints and printing inks (International Programme
on Chemical Safety, 1995; Samimi, 1982). While it has been detected at low levels in foods (Sasaki et al., 2005; Kataoka et





al., 2007), it is not detected in food cooking emissions (Rogge et al., 1991; Schauer et al., 1999a, 2002a), so isophorone likely arises from a separate and temporally coincident source.

While little information is available about sources of 1-tridecene specifically, lower molecular weight alkenes come from mostly anthropogenic origin in urban areas, specifically mobile sources (Luecken et al., 2012). In this data set, 1-
tridecene correlates better with diesel tracers (r ≈ 0.7) than gasoline tracers (r ≈ 0.55), though it is not specifically mentioned in speciated diesel composition studies (Rogge et al., 1993; Schauer et al., 1999b; Gentner et al., 2013). Even at its peak, the 1-tridecene concentration is less than 4 ng m$^{-3}$ (0.5 ppt).

### 3.13 Factor 12: Isoprene

Factor 12 contains 73 % of the mass of isoprene, and no more than 10 % of the mass of any other compound. The
concentration is elevated between the early morning hours and late evening hours, with a peak in the early evening (18:00 LT) and a much smaller peak in the morning (04:00 to 06:00 LT). When the concentration of this factor is high, wind speeds are typically elevated and from the west (Fig. 5).

Isoprene is predominantly of biogenic origin and its emission is light- and temperature-dependent (Guenther, 1997). Seasonal output varies greatly, with maximum emission in the summer months (Palmer et al., 2006; Liakakou et al., 2007).
Summertime diurnal concentration profiles of isoprene typically increase from morning to a midday maximum, declining again by nightfall, in accordance with its sensitivity to light and temperature. In wintertime when biogenic production is low, isoprene has been observed to correlate with pollutants of known vehicle traffic origin in urban areas and is inferred to originate from that source (Reimann et al., 2000; Borbon et al., 2001; Lee and Wang, 2006; Hellén et al., 2012; Kaltsonoudis et al., 2016).

As spring represents a transition from winter to summer conditions, evidence of both anthropogenic and biogenic isoprene is present in Livermore. The small early morning peak is unlikely to be of biogenic origin since it occurs when temperatures are coldest and before the sun rises. The peak coincides with those of the traffic markers; restricted to the hours of 22:00 to 05:30 LT, isoprene correlates with benzene and several other gasoline markers (methylcyclopentane through p-diethylbenzene in Table 1 and Fig. 4). The correlation between benzene and isoprene is shown in Fig. 8. A linear regression
of the data puts the intercept within error of the origin, suggesting no additional sources for isoprene or benzene overnight. The slope, which represents an average ratio, is 0.22. Analysis of vehicle emissions measured during dynamometer tests performed in 2014 (Drozd et al., 2016) show a median isoprene:benzene ratio of 0.18 with 50 % of the data between 0.14 – 0.34 (Fig. S7), providing independent support for attributing these isoprene emissions to anthropogenic sources.

In contrast, the late afternoon or early evening peaks roughly correlate with the maximum afternoon temperature,
though the peaks occur a few hours after the temperature maximum. Midday concentrations may be depressed by reaction with the hydroxyl radical as well as dilution from the heightened boundary layer and high winds. In the early evening there is a narrow window of time when the sun has set (and therefore hydroxyl radical production has ceased), but isoprene emission continues. The lowering boundary layer and lower wind speeds may also contribute to the heightened isoprene





concentrations. Later in the evening, emission ceases and isoprene is rapidly lost to reaction with nitrate radical and ozone

(Steinbacher et al., 2005).

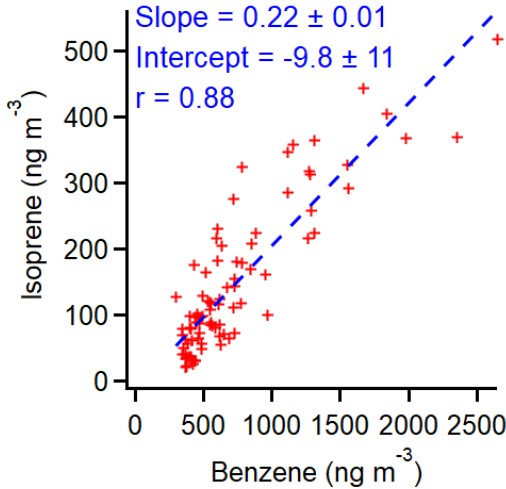

**Figure 8. Correlation of isoprene and benzene between 22:00 and 05:30 LT, when biogenic production of isoprene is expected to be negligible.**

**3.14 Factor 13: Possible jet fuel event**

Factor 13 captures a short episode of heightened gasoline tracer concentrations at 23:00 on 13 April. It is not associated with a sudden shift in wind patterns, but more likely represents a transient source. Around 20 % of the mass of $C_6$–$C_{10}$ linear, branched and aromatic hydrocarbons is included in this factor, with smaller contributions out to $C_{14}$. Compositionally, Factor 13 differs from Factor 3 (gasoline) in its lack of other compound classes; Factor 3 additionally

contains contributions from PAHs, palmitoleic acid and D4 and D5 siloxanes. Since D5 originates from personal care products likely applied in the morning hours, its absence in Factor 13 is not surprising. Similarly, as mentioned in the Factor 3 discussion, palmitoleic acid and D4 coincide temporally with regular morning gasoline emissions but do not come from gasoline, so are not expected in Factor 13 either. The PAH presence in Factor 3 was ascribed to gasoline exhaust, so its absence in this factor is inconsistent with a gasoline source. One possibility is that Factor 13 captures a low jet aircraft

flyover, since the carbon number range of jet fuel approximately matches the factor's chemical profile (Masiol and Harrison, 2014) but jet fuel is depleted in PAHs compared to gasoline (Shumway, 2000). The likely origin or destination of the aircraft is Livermore Municipal Airport, located approximately 3 km away from the sampling site.





## 4 Conclusion

For the first time on a single instrument, hourly measurements were successfully carried out of over 160 organic
compounds in the ambient air ranging from $C_5$ to $C_{27}$ alkane equivalent volatility in both gas and particle phases.
Measurements took place in Livermore, California between 11 April and 21 April 2018 with the cTAG. PMF was applied to
a subset of the measured compounds to elucidate the major sources of pollution in Livermore in the springtime.

Major factors observed to contribute significantly included gasoline emissions, consumer product emissions,
emissions from evaporative (temperature-dependent) sources, biomass burning, secondary oxidation, background emissions
from continuously emitting sources or long-lived pollutants, and several factors associated with single compounds or specific
events during the campaign described by a unique compositional profile. The gasoline factor had a morning maximum, but
lacked the evening enhancement observed in other studies, likely due to the residential area of our sampling location and
emissions from modern vehicles being concentrated in cold starts. Monoterpenoid compounds were associated with the
personal care product factor more than any other factor, highlighting the dominance in urbanized areas of anthropogenic
sources for some compounds normally from biogenic sources. No clear diesel factor emerged; rather, diesel tracers were
split between three different factors primarily associated with distinct single events. Isoprene's dual biogenic and vehicle
exhaust sources combined with atmospheric chemistry suppressing midday concentrations forced it into its own factor.
While PCBTF and isophorone are both industrial chemicals associated with solvents, they exhibit extremely distinct
temporal profiles and are present in different factors, suggesting distinct sources. The clearly separated biomass burning
factor demonstrates that residential wood burning is still an important source of organic emissions even in the springtime.

This analysis underscores the increasing importance of anthropogenic petroleum-derived VOCs from non-mobile
sources in a suburban environment, an emerging topic of interest in recent years (McDonald et al., 2018). The ability to
resolve individual isomers at high time resolution proved crucial, as it allowed for (1) the separation of the monoterpenes
between consumer product emissions and a suspected biogenic source, and (2) the distinct categorization and interpretation
of the nitrophenol and dichlorobenzene isomers. Including VOCs, IVOCs and SVOCs together in a single analysis expanded
the profiles of some sources dominated by VOCs, such as the IVOCs methyl salicylate and α-isomethyl ionone being
included in the consumer product factor. Similarly, the secondary oxidation/persistent personal care product emissions factor
included a mix of SVOCs, stable VOCs and oxygenated VOCs, constituting a unique profile distinct from fresh personal
care product emissions. Measurement of compounds over a wide range of volatilities and oxidation states can allow for more
detailed source characterization and tracking of atmospheric processes than focusing on VOCs or particulate matter alone.

*Data availability.* Concentration timelines, positive matrix factorization results, and data for each figure are available from
the first author upon request.





*Author Contributions.* RAW, NMK, RJW and AHG designed and coordinated the Livermore 2018 deployment and RAW and NMK collected data from that deployment. RAW and GTD performed data analysis. RAW prepared the manuscript with input and revisions from NMK, GTD and AHG.

*Competing Interests.* The authors declare that they have no conflict of interest.


*Disclaimer.* This publication has not been formally reviewed by the EPA. The views expressed in this publication are solely those of the authors, and the EPA does not endorse any products or commercial services mentioned in this publication.

*Acknowledgments.* The Livermore Area Recreation and Park District (LARPD) provided the site and infrastructure in
Livermore for the spring 2018 field deployment. The authors would like to especially thank Bruce Aizawa and Steve Sommers of LARPD for enabling us to do measurements at the site. Hourly temperature, wind speed and direction, and ozone measured approximately 100 m away from our site were provided by the Bay Area Air Quality Management District.

*Financial support.* cTAG development, including the field deployment in Livermore, was supported by the US Department
of Energy SBIR/STTR under grant DE-SC0011397. Rebecca A. Wernis was supported by the U.S. Environmental Protection Agency (EPA) Science To Achieve Results (STAR) fellowship assistance agreement no. FP-91778401-0.

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

U.S. EPA (Environmental Protection Agency) Comptox Chemicals Dashboard: 2-Methoxynaphthalene: https://comptox.epa.gov/dashboard/chemical/details/DTXSID7044392, last access: 12 April 2022.

U.S. EPA (Environmental Protection Agency) Comptox Chemicals Dashboard: 2-Nitrophenol: https://comptox.epa.gov/dashboard/chemical/details/DTXSID1021790, last access: 5 April 2022.

U.S. EPA (Environmental Protection Agency) Comptox Chemicals Dashboard: 4-Hydroxybenzoic Acid: https://comptox.epa.gov/dashboard/chemical/details/DTXSID3026647, last access: 12 April 2022.

U.S. EPA (Environmental Protection Agency) Comptox Chemicals Dashboard: 4-Nitrophenol: https://comptox.epa.gov/dashboard/chemical/details/DTXSID0021834, last access: 5 April 2022.

U.S. EPA (Environmental Protection Agency) Comptox Chemicals Dashboard: Benzophenone: https://comptox.epa.gov/dashboard/chemical/details/DTXSID0021961, last access: 28 March 2022.

U.S. EPA (Environmental Protection Agency) Comptox Chemicals Dashboard: p-Anisic Acid: https://comptox.epa.gov/dashboard/chemical/details/DTXSID4059205, last access: 14 April 2022.

U.S. EPA (Environmental Protection Agency) Comptox Chemicals Dashboard: Phthalimide: https://comptox.epa.gov/dashboard/chemical/details/DTXSID3026514, last access: 12 April 2022.

Vione, D., Maurino, V., Minero, C., Duncianu, M., Olariu, R.-I., Arsene, C., Sarakha, M., and Mailhot, G.: Assessing the transformation kinetics of 2- and 4-nitrophenol in the atmospheric aqueous phase. Implications for the distribution of both nitroisomers in the atmosphere, Atmos. Environ., 43, 2321–2327, https://doi.org/10.1016/j.atmosenv.2009.01.025, 2009.

Wang, H., Gao, Y., Wang, S., Wu, X., Liu, Y., Li, X., Huang, D., Lou, S., Wu, Z., Guo, S., Jing, S., Li, Y., Huang, C., Tyndall, G. S., Orlando, J. J., and Zhang, X.: Atmospheric Processing of Nitrophenols and Nitrocresols From Biomass Burning Emissions, J. Geophys. Res. Atmos., 125, e2020JD033401, https://doi.org/10.1029/2020JD033401, 2020.

Wang, L., Atkinson, R., and Arey, J.: Dicarbonyl Products of the OH Radical-Initiated Reactions of Naphthalene and the C1- and C2-Alkylnaphthalenes, Environ. Sci. Technol., 41, 2803–2810, https://doi.org/10.1021/es0628102, 2007.





Wang, L., Wang, X., Gu, R., Wang, H., Yao, L., Wen, L., Zhu, F., Wang, W., Xue, L., Yang, L., Lu, K., Chen, J., Wang, T., Zhang, Y., and Wang, W.: Observations of fine particulate nitrated phenols in four sites in northern China: concentrations, source apportionment, and secondary formation, Atmos. Chem. Phys., 18, 4349–4359, https://doi.org/10.5194/acp-18-4349-2018, 2018.

Wang, Q., Huang, X. H. H., Tam, F. C. V., Zhang, X., Liu, K. M., Yeung, C., Feng, Y., Cheng, Y. Y., Wong, Y. K., Ng, W. M., Wu, C., Zhang, Q., Zhang, T., Lau, N. T., Yuan, Z., Lau, A. K. H., and Yu, J. Z.: Source apportionment of fine particulate matter in Macao, China with and without organic tracers: A comparative study using positive matrix factorization, Atmos. Environ., 198, 183–193, https://doi.org/10.1016/j.atmosenv.2018.10.057, 2019a.

Wang, R., Moody, R., Koniecki, D., and Zhu, J.: Low molecular weight cyclic volatile methylsiloxanes in cosmetic products sold in Canada: implication for dermal exposure., Environ Int, 35, 900–904, https://doi.org/10.1016/j.envint.2009.03.009, 2009.

Wang, X., Gu, R., Wang, L., Xu, W., Zhang, Y., Chen, B., Li, W., Xue, L., Chen, J., and Wang, W.: Emissions of fine particulate nitrated phenols from the burning of five common types of biomass, Environ. Pollut., 230, 405–412, https://doi.org/10.1016/j.envpol.2017.06.072, 2017.

Wang, Y., Hu, M., Wang, Y., Zheng, J., Shang, D., Yang, Y., Liu, Y., Li, X., Tang, R., Zhu, W., Du, Z., Wu, Y., Guo, S., Wu, Z., Lou, S., Hallquist, M., and Yu, J. Z.: The formation of nitro-aromatic compounds under high $NO_x$ and anthropogenic VOC conditions in urban Beijing, China, Atmos. Chem. Phys., 19, 7649–7665, https://doi.org/10.5194/acp-19-7649-2019, 2019b.

Warneke, C., de Gouw, J. A., Holloway, J. S., Peischl, J., Ryerson, T. B., Atlas, E., Blake, D., Trainer, M., and Parrish, D. D.: Multiyear trends in volatile organic compounds in Los Angeles, California: Five decades of decreasing emissions, J. Geophys. Res. Atmos., 117, https://doi.org/10.1029/2012JD017899, 2012.

Weitkamp, E. A., Sage, A. M., Pierce, J. R., Donahue, N. M., and Robinson, A. L.: Organic Aerosol Formation from Photochemical Oxidation of Diesel Exhaust in a Smog Chamber, Environ. Sci. Technol., 41, 6969–6975, https://doi.org/10.1021/es070193r, 2007.

Wernis, R. A., Kreisberg, N. M., Weber, R. J., Liang, Y., Jayne, J., Hering, S., and Goldstein, A. H.: Development of an in situ dual-channel thermal desorption gas chromatography instrument for consistent quantification of volatile, intermediate-volatility and semivolatile organic compounds, Atmos. Meas. Tech., 14, 6533–6550, https://doi.org/10.5194/amt-14-6533-2021, 2021.

Westerling, A. L.: Increasing western US forest wildfire activity: sensitivity to changes in the timing of spring, Phil. Trans. R. Soc. B, 371, 20150178, https://doi.org/10.1098/rstb.2015.0178, 2016.

Westerlund, J., Bryngelsson, I.-L., Löfstedt, H., Eriksson, K., Westberg, H., and Graff, P.: Occupational exposure to trichloramine and trihalomethanes: adverse health effects among personnel in habilitation and rehabilitation swimming pools, J. Occup. Environ. Hyg., 16, 78–88, https://doi.org/10.1080/15459624.2018.1536825, 2019.





World Meteorological Organization, United States, National Oceanic and Atmospheric Administration, United States, National Aeronautics and Space Administration, United Nations Environment Programme, and European Commission: Scientific assessment of ozone depletion: 2018., 2019.

Yao, D., Lyu, X., Lu, H., Zeng, L., Liu, T., Chan, C. K., and Guo, H.: Characteristics, sources and evolution processes of atmospheric organic aerosols at a roadside site in Hong Kong, Atmos. Environ., 252, 118298, https://doi.org/10.1016/j.atmosenv.2021.118298, 2021.

Yuan, B., Shao, M., de Gouw, J., Parrish, D. D., Lu, S., Wang, M., Zeng, L., Zhang, Q., Song, Y., Zhang, J., and Hu, M.: Volatile organic compounds (VOCs) in urban air: How chemistry affects the interpretation of positive matrix factorization

(PMF) analysis, J. Geophys. Res. Atmos., 117, https://doi.org/10.1029/2012JD018236, 2012.

Yuan, B., Liggio, J., Wentzell, J., Li, S.-M., Stark, H., Roberts, J. M., Gilman, J., Lerner, B., Warneke, C., Li, R., Leithead, A., Osthoff, H. D., Wild, R., Brown, S. S., and de Gouw, J. A.: Secondary formation of nitrated phenols: insights from observations during the Uintah Basin Winter Ozone Study (UBWOS) 2014, Atmos. Chem. Phys., 16, 2139–2153, https://doi.org/10.5194/acp-16-2139-2016, 2016.

Yuan, Z., Lau, A. K. H., Shao, M., Louie, P. K. K., Liu, S. C., and Zhu, T.: Source analysis of volatile organic compounds by positive matrix factorization in urban and rural environments in Beijing, J. Geophys. Res. Atmos., 114, https://doi.org/10.1029/2008JD011190, 2009.

Zota, A. R., Calafat, A. M., and Woodruff, T. J.: Temporal Trends in Phthalate Exposures: Findings from the National Health and Nutrition Examination Survey, 2001–2010, Environ. Health Perspect., 122, 235–241,

https://doi.org/10.1289/ehp.1306681, 2014.

Zwiener, C., Richardson, S. D., De Marini, D. M., Grummt, T., Glauner, T., and Frimmel, F. H.: Drowning in Disinfection Byproducts? Assessing Swimming Pool Water, Environ. Sci. Technol., 41, 363–372, https://doi.org/10.1021/es062367v, 2007.