# Peer review of "Source Apportionment of VOCs, IVOCs, and SVOCs by Positive Matrix Factorization in Suburban Livermore, California"

_EGUsphere, 2022_

## Author Comment (AC1)

**1ˢᵗ Reviewer's Comments With Inline Responses**

The author deployed a state-of-the-art instrument named cTAG to simultaneously measure speciated VOCs, IVOCs, and SVOCs. Such instrument is a powerful tool to study atmospheric organic carbon, given its wide range in volatility and degree of oxidation in the atmosphere. The sources of measured compounds were apportioned using the PMF model, and 13 factors were finally resolved. The results described comprehensive and detailed sources of organics in the real world (e.g., personal care products and asphalt emissions), and also the secondary oxidation processes. The measurement and calibration appear to be carefully performed, and the paper is clearly written. In reading through the preprint, it is considered that the article still needs to address the following concerns:

The sources of organic carbon might be closely associated with the sampling site and surrounding environments. I suggest the author provided a map to describe the geographical location of the sampling site, which is also helpful for readers to better understand the source interpretation in the following context.

**We have added a map as suggested. It is Figure 1 in the revised manuscript.**

Hourly VOC, IVOC, and SVOC speciated measurement was designed for this study. Compared with I/VOC channel, the SVOC analysis needs online derivatization. I recall in the previous AMT paper (Wernis et al., 2021), the authors measured SVOCs in a bihourly cycle. I wonder if the reduced time resolution affects the derivation efficiency.

**Sorry if this was not clear in the previous AMT paper: cTAG has always measured SVOCs hourly, but with alternating particle-only and gas-plus-particle measurement (hence bihourly resolution for each of these types of measurements). The sample has always been ~23 minutes long and has had online derivatization applied, so there are no variations in time resolution that could affect derivatization efficiency. In this manuscript, as reviewer 1 states, there are hourly measurements of VOCs, IVOCs and SVOCs, but for the PMF analysis only measurements from every other hour were used in alignment with the gas-plus-particle SVOC measurements, as discussed in section 2.3.**

Some problems are related to PMF analysis.

I noticed that the author put much effort into interpreting the source profile for individual sources. However, the contribution of individuals should also be presented, which is helpful to evaluate the effects of the individual source on atmospheric organic carbon abundance. I made a rough estimation for the individual source contribution based on Figure 3. It seems that Factor 3 (gasoline) and Factor 5 (consumer) should be the two largest contributors. However, the represented compounds in these two factors are most I/VOCs. As the author has mentioned in line 268 and shown in Figure S6, the I/VOCs compounds dominated the total measured compounds (87%) and resolved source profile. It is not surprising that the fraction of compounds of high volatility will be dominated all measured compounds across wide volatility, as they are more easily dispersed in the atmosphere. Therefore, my major concern is if you input species across such wide range of volatility into PMF, will the high

mass fraction of high-volatility species also dominate the apportioned source? In other words, if you only use the SVOCs dataset, will some new sources be resolved?

**It is true that overall I/VOCs contribute considerably greater mass and thus may exert a greater influence on the overall factor timelines. Our analysis includes compounds over the entire volatility range (VOC, IVOC and SVOC) in order to discover and chemically resolve sources that contain both I/VOCs and SVOCs, which an SVOC-only analysis would not achieve. While an SVOC-only PMF analysis would be of great interest and would complement this analysis, it is outside the scope of the current manuscript.**

Factor 1 represents the long-lived and continuously emitted. If the factor is continuously emitted, why did the peak value of this factor occur at night (Figure 3)? Moreover, for the cooking/diesel factor (Factor 11), why the peak value was also found at night? To my understanding, this factor should be associated with human-related activities. It would be helpful if the author could clarify these trivial issues in the manuscript.

**Regarding Factor 1, the diurnal plot indicates large variability at each hour, indicating the nighttime enhancement is weak. We believe this factor is primarily capturing compounds present at all hours of the day, though slight variability in the concentration of each compound from hour to hour leads to an incidental overall nighttime enhancement. To clarify this, we have modified a sentence in the first paragraph of section 3.2 from:**

**"This factor is mildly elevated at night, but with high uncertainty, and compounds with a slightly opposite diurnal profile (e.g. 4-hydroxybenzoic acid, phthalimide) can also be found in this factor."**

**to**

**"The diurnal profile of this factor (Fig. 4(b)) indicates mild nighttime concentration enhancement, but with large variability, and compounds with a slightly opposite diurnal profile (e.g. 4-hydroxybenzoic acid, phthalimide) can also be found in this factor."**

**For Factor 11, we infer that the source of the spike in concentration of cooking emissions is coming from a nearby commercial kitchen, which could be getting an early start (4AM local daylight time) to food preparation for the day. We have added a sentence to the second paragraph of section 3.12 clarifying this:**

**"Residential cooking is an unlikely explanation given the early time of the event (04:00 local daylight time), but commercial cooking preparations could plausibly begin at such an early hour."**

**A clear residential cooking factor did not emerge from this analysis.**

Figure 4: the association of arrow length and wind speed should be added as a legend on the graph. Besides, it would be more helpful if the author could add some dashed lines for the readers to separate each day.

**The suggested modifications have been made.**

**2nd Reviewer's Comments with Inline Responses**

Wernis et al. present a comprehensive source apportionment of atmospheric VOCs, IVOCs and SVOCs for an Eastern US suburban area, which can be considered representative enough to provide insights into the various sources of complex atmospheric organic carbon in such settings. They deploy a novel instrument in order to access a broad volatility range in their source apportionment which is considered a real step forward. The combined gas chromatographic and mass spectrometric approach provides unique capabilities of compound identification, and the authors take great care in analyzing their PMF-obtained variability in the data. This manuscript shows that in suburban areas, organic compounds associated with personal care products are present in many ways. In addition, it also shows that classical source profiles such as gasoline emissions can have distinct features in such purely suburban areas. In that sense this manuscript clearly improves our understanding of the diverse sources of organic compounds in the atmosphere.

The manuscript is well-written and the scientific approach seems solid to me. I especially like how the authors took great care in differentiating between source apportionment and possible lifetime differences in their discussion of the different factors. However, I find it currently very cumbersome to go through the manuscript and relate the identification of the different PMF factors with the shown Figures and Tables. In addition, I would also appreciate a more thorough discussion on the uncertainty of the individual factor profiles, potentially explaining some of the "cross-talk" between different factors. Therefore, I hope that the following comments can help to improve the clarity of the manuscript and help to better streamline the main messages.

Major comments:

I find it very cumbersome to swap between the different plots (Fig.1-3, Fig. 5) during the discussion of each factor and it currently is a lot of work to thoroughly go through the manuscript. The following points should be improved:

In my opinion, each factor should be associated with a single figure showing the time-series, the diurnal variation and the factor profile. In return, Fig.5, Fig. 3 and Fig.1 (anyways roughly shown in Fig. 4 c) could be removed or put into the SI.

**Thank you for the suggestion. We have created individual figures for each factor incorporating the corresponding plots from Figs. 1, 2, 3 and 5. We have kept Fig. 2 in the main text for ease of comparison across factors, but have moved Figs. 1, 3 and 5 to the supplement.**

For the factor profile (Fig 2.) it would be nice to add the names of the most prominent peaks, or at the very least mention the number of the compound as given in Table 1 in the text. This would help the reader to much faster connect the compound to the profile.

**Thank you for this suggestion. We have added the compound index to the top of the figure in addition to the bottom to facilitate identifying the individual bars in the plot. We feel that adding names or indices of prominent peaks would make an already very dense figure too cluttered.**

In that sense, I would appreciate if during the discussion of the factors also the relative contributions of different compounds to that factor compared to other compounds (maybe call it relative contribution) could be mentioned and not only the relative contribution of a compound to that factor compared to their contribution to other factors (maybe call it relative split) would be mentioned. This should provide insights into which compounds are the most dominant one. Subsequently the authors should be careful with their wording when using phrases such as "minor contribution" to make clear which type of contribution they mean.

**Thank you for this suggestion. We agree that the compounds that contribute the most mass to a given factor (i.e. highest relative contribution) are of interest, and are not always the same set of compounds that contribute the greatest fraction of their total mass to a given factor (i.e. highest relative split). Therefore we have added a table for each factor in the supplement indicating the top mass contributors to each factor. However, due to the often orders of magnitude differences in concentration between compounds, especially between I/VOCs and SVOCs, and due to the fact that no attempt was made to account for all of the mass of VOCs, IVOCs and SVOCs, we believe the compounds with the highest relative split offer clearer insight into factor interpretation than compounds with the highest relative contribution. We have added a paragraph to the end of section 3.1 to clarify this:**

**"Since uncertainty generally scales with concentration, even the error in the contribution of a high mass VOC such as chloroform could be greater than the contribution from the most prominent SVOCs, making drawing conclusions from the mass composition of factors challenging. Additionally, this analysis is not a comprehensive account of all the organic carbon in the measured volatility range; there could be compounds that were not quantified (and thus were excluded from this analysis) that would contribute significant mass to the determined factors. For these reasons, our factor interpretations are mainly based on which compounds have the greatest portion of their mass in each factor ("mass fraction"), rather than which compounds contribute the most mass. The latter information is included in Sect. S5 for interested readers."**

For Figure 2 (i.e. the subplots in the future factor specific Figures), I would suggest that the x-axis gets some coloring where IVOC/VOC compounds are and where SVOC compounds are. This would help to better assign the contributions of the different modes of the cTAG to the factors and reduce the need of the overly lengthy Table 1, which I would suggest moving to the SI to keep the main text better accessible.

**We have added x-axis coloring to differentiate between compounds measured on the I/VOC and SVOC channels. Since Table 1 is the only way to link the compound indices used in Figure 2 to compound names, we prefer to keep it in the main text.**

Please check careful that all axes have units. Fig. 5 is really difficult to put into context without a map of Livermore and its surroundings and the location of the measurement station indicated on it. I really suggest the authors to add such a map.

**We have added number labels to the rings of the roses in Fig. 5 (now spread out in sub-figures for each factor). We have also added a map as suggested. It is Figure 1 in the revised manuscript.**

When discussing the factor profiles, the authors find several compounds showing up in factors where they wouldn't be primarily attributed to. I appreciate the authors discussion on these points, but I miss an

overall error estimate of the individual source profiles, which might explain some of the interesting assignments (e.g., siloxanes and palmitoleic acid in Factor 3 and diesel markers in the primary biogenic factor). If the authors can use bootstrapping on their dataset (removing or doubling randomly datapoints and rerunning PMF), this should allow them to assign errorbars to the individual contributions given in Figure 2, which could indicate that some assignments are more uncertain than others (as the authors already explain for example in their discussion about palmitoleic acid in Factor 3).

**Thank you for this suggestion. We have added 5th and 95th percentile bootstrapping values as estimates of uncertainty to Figure 2 and updated the figure caption accordingly. We have also added a section in the supplement briefly describing bootstrapping for uncertainty estimation:**

**"S2.3 Factor profile uncertainty**

> **Bootstrapping analysis performs PMF on various resampled versions of the original data set. The factors derived in each bootstrap solution are mapped to the original solution based on their correlation with each of the factors in the original solution. Repeated bootstrapping yields a distribution of values for each compound's fractional contribution to each factor, a measure of the uncertainty of the original solution. 5th and 95th percentile values from this analysis are shown in Fig. 2 as an uncertainty estimate. A more detailed description of bootstrapping analysis as a method to estimate uncertainty can be found in Paatero et al. (2014) and Brown et al. (2015)."**

**The uncertainty estimate for palmitoleic acid is higher than most other compounds, as expected. We have added a sentence to the end of the third paragraph of section 3.4 explicitly pointing this out (underlined):**

**"Factors 5 and 6 and many marker compounds share this general pattern of elevated concentrations exclusively in the early morning hours. They are governed by the same large-scale atmospheric processes. Thus while differences exist which allow them to be separated by the PMF model, which will be discussed in the descriptions for those factors, their overarching similarity means the sources are not separated perfectly. For example, palmitoleic acid is primarily a tracer of cooking (Rogge et al., 1996; Robinson et al., 2006), yet has its highest mass fraction in this factor, while other cooking tracers appear in other factors. Together with the overall similarity in variability between gasoline and cooking markers, palmitoleic acid's low and noisy signal may have caused it to be placed in this factor primarily by the model, even though it does not originate from the same pollution source. The high uncertainty of palmitoleic acid's allocation between factors 3, 5, 11 and 13 (compound 99 in Fig. 2) confirms this."**

Minor comments:

   Page 3, lines 94-95: It would help the reader to directly give the corresponding saturation mass concentrations to the given alkene equivalents.

**We have changed the lines in question to read:**

**"This 10.0 L min$^{-1}$ is then pulled through a coated metal mesh filter cell held at 30 °C which collects IVOCs and SVOCs between $C_{14}$ and $C_{32}$ alkane equivalent volatility ($C^* \approx 10^{-1}$ to $10^5$ μg m$^{-3}$). The remaining 100 sccm is pulled through a bed of adsorbent materials also held at 30 °C designed to efficiently collect VOCs and IVOCs between $C_5$ and $C_{16}$ alkane equivalent volatility ($C^* \approx 10^5$ to $10^{10}$ μg m$^{-3}$)."**

Page 4, line 111-112: Quickly recall for the reader what you mean by total ion chromatograms (total signal in mass spectrometer) and especially single ion chromatograms (specific mass identified by the mass spectrometer).

**We have changed the lines in question to read:**

**"Two chromatograms with mass spectral information are generated every hour – one for I/VOCs and one for 110 SVOCs. Typical total ion chromatograms (TICs), which include the signal from all mass-to-charge ratios added together, can contain hundreds to thousands of compounds, often leading to overlapping peaks. Single ion chromatograms (SICs) consist of the signal from a single mass-to-charge ratio and have far fewer overlaps; thus integrated peaks on the SIC of a prominent mass-to-charge ratio in the target compound's mass spectrum are used as the basis for quantification."**

Page 8, line 235: Qexp is not defined in the main text yet, so it would be good to refer to the SI here once again.

**We have changed the line in question to read:**

**"Solutions with fewer factors fail to separate factors with meaningful physical interpretations and do not incorporate one of the largest reductions in $Q/Q_{exp}$ (defined in Sect. S2.1)."**

Page 14, line 276: Do I understand correctly, that the percentages in brackets give the contribution of the individual compound to that factor compared to its contribution to other factors? Please specify what these numbers mean (see major comment)

**Yes, that is correct. We have reworded the second sentence of section 3.2 to clarify this:**

**"Chloroform and bromoform have the highest fraction of their mass in this factor compared to other factors (33 % and 28 % respectively; this quantity for each compound is hereafter referred to as "mass fraction"). Other well-represented compounds include 4-hydroxybenzoic acid (mass fraction 35 %), p-anisic acid (26 %), phthalimide (37 %), 2-cyclopenten-1-one (32 %), and 2-methoxynaphthalene (33 %)."**

Page 15, line 283 and line 304: Either I have difficulties in understanding the rose plot, or that southwest enhancement does not look very mild to my eyes. Please consider revising these statements. Also for other factors I am not convinced if a few occurrences from other directions outweigh a dominant occurrence from one direction to conclude that the factor has no or mild directional dependence.

**Regarding Factor 1, we have modified the relevant text from:**

**Line 283: "When Factor 1 is elevated winds come from all directions, with a mild southwest preference (Fig. 5)."**

**Line 304: "Because Factor 1 is associated with winds from all directions, the sources would need to be either hyper local or ubiquitous in the surrounding area. Agricultural activity may be able to account for 4-hydroxybenzoic acid, p-anisic acid, phthalimide, and 2-cyclopenten-1-one."**

**to**

**Line 283: "When Factor 1 is elevated winds come predominantly from the southwest (Fig. 5)."**

**Line 304: "Agricultural activity may be able to account for 4-hydroxybenzoic acid, p-anisic acid, phthalimide, and 2-cyclopenten-1-one, perhaps from the vineyards approximately 4 to 5 km to the south of the sampling site."**

**We have also adjusted the language in the first paragraph of the Factor 9 description from:**

**"Winds are light (< 1.5 m s$^{-1}$ except for one data point) and come from all directions, with a mild northeast preference, when Factor 9 is elevated (Fig. 5)."**

**to**

**"Winds are light (< 1.5 m s$^{-1}$ except for one data point) and come from all directions, but predominantly from the northeast, when Factor 9 is elevated (Fig. 14(c))."**

Page 22, line 477: The very early morning peak of Factor 5 puzzles me, as I do not believe that typical wake-up time (commuter preparing for work which should induce the personal care product emissions) in Livermore is 4 am. However, as noted later, it is consistently a bit earlier than the gasoline factor, which is very plausible. Maybe there is some daylight-saving-time related shift, which could explain both peaks to be so early in the day.

**Indeed daylight savings time plays a role here. We have added language to clarify this:**

**Line 355 (section 3.4): "This factor exhibits a strong diurnal pattern with sharp peaks in concentration in the early morning (between 05:00 and 07:00 local standard time, or 06:00 and 08:00 daylight time, which was in effect during the measurement period) and near zero loading at other times of day."**

**Line 477: "Peak consumer products emissions occur slightly earlier than gasoline emissions on average, from about 04:00 to 06:00 local standard time, or 05:00 to 07:00 local daylight time (Fig. 3)."**

Page 22, line 496: "Methyl salicylate and α-isomethyl ionone are low-volatility IVOCs, measured on the SVOC channel" seems to be a direct contradiction. A compound which is an IVOC cannot be of low-volatility, as it would be an LVOC then (not even SVOC). Moreover, why is it an IVOC if measured in the SVOC channel? Please revise this statements and clarify.

**We have changed the line in question to read:**

**"Methyl salicylate and α-isomethyl ionone are relatively low-volatility IVOCs, measured on the SVOC channel of cTAG."**

**Our intent was to convey that within the range of volatility encompassed by IVOCs, these two compounds are on the lower volatility end. The SVOC channel predominantly collects SVOCs, but its collection range includes a small portion of IVOCs on the lower volatility side, as stated in line 94.**

**List of Changes in Manuscript**

[revised manuscript text omitted]

- Pg 15, Ln 295 etc. Added Figures 4, 5, 6, 7, 10, 11, 12, 13, 14, 15, 16, 17, and 19, which contain the timeline, diurnal profile, wind direction information and factor composition profile for each of the 13 factors presented.
- Pg 15, Ln 319 Removed "Because Factor 1 is associated with winds from all directions, the sources would need to be either hyper local or ubiquitous in the surrounding area"
- Pg 15, Ln 320 Added "…, perhaps from the vineyards approximately 4 to 5 km to the south of the sampling site."
- Pg 18, Ln 371 added "…,or 06:00 and 08:00 local daylight time, which was in effect during the measurement period"
- Pg 19, Ln 395 Added "The high uncertainty of palmitoleic acid's allocation between factors 3, 5, 11 and 13 (compound 99 in Fig. 2) confirms this."
- Pg 21, Ln 461 Changed colors in Figure 8 (formerly Figure 6) to be more colorblind friendly.
- Pg 22, Ln 470 Added dashes and dots to red trace in Figure 9 (formerly Figure 7) to be more colorblind friendly.
- Pg 23, Ln 498 Removed "LT" and added "local standard time, or 05:00 to 07:00 local daylight time"
- Pg 24, Ln 520 added "relatively"
- Pg 29, Ln 641 changed "…with a mild northeast preference…" to "…but predominantly from the northeast…"
- Pg 32, Ln 700 added "Residential cooking is an unlikely explanation given the early time of the event (04:00 local daylight time), but commercial cooking preparations could plausibly begin at such an early hour."
- Updated figure references throughout manuscript.

Supplement changes

- Pg 2, Ln 31 Changed color scheme of Figure S1 to make figure more colorblind friendly.
- Pg 4, Ln 66 Added Section S2.3:
  - "S2.3 Factor profile uncertainty
  - Bootstrapping analysis performs PMF on various resampled versions of the original data set. The factors derived in each bootstrap solution are mapped to the original solution based on their correlation with each of the factors in the original solution. Repeated bootstrapping yields a distribution of values for each compound's fractional contribution to each factor, a measure of the uncertainty of the original solution. 5th and 95th percentile

values from this analysis are shown in Fig. 2 and part (d) of Figs. 4-7, 10-17 and 19 as an uncertainty estimate. A more detailed description of bootstrapping analysis as a method to estimate uncertainty can be found in Paatero et al. (2014) and Brown et al. (2015)."

- Pg 6, Ln 83 added section S3, "13 factor solute figures for comparison across factors, containing figures S6, S7 and S8 (formerly Figs 1, 3 and 5).
- Pg 8, Ln 91 added number markers to rings of rose plots in Figure S8 (formerly Figure 5)
- Pg 9, Ln 101 Added Section S5:
  o "S5 Top mass contributors to each factor
  o Tables S2-S14 display the top 10 compounds contributing the most mass to each factor. This is distinct from the compounds that have the greatest fraction of their mass in each factor, which are presented in Fig. 2 and subfigure (d) of Figs. 4-7, 10-17 and 19 and are discussed in detail in the main text. The fraction of each compound's mass in each factor ("mass fraction") for the top 10 mass contributors is also presented in Tables S2-S14. For example, 43% of the mass of Factor 1 is chloroform, while 33% of chloroform's mass is attributed to Factor 1."
- Pg 10, Ln 114 Added Tables S2 through S14, listing the top 10 mass contributors to each factor and their fractional contributions.